# Observations of diapycnal upwelling within a sloping submarine canyon

Bethan L. Wynne-Cattanach[1✉], Nicole Couto[1], Henri F. Drake[2], Raffaele Ferrari[3], Arnaud Le Boyer[1], Herlé Mercier[4], Marie-José Messias[5], Xiaozhou Ruan[6], Carl P. Spingys[7], Hans van Haren[8], Gunnar Voet[1], Kurt Polzin[9], Alberto C. Naveira Garabato[10] & Matthew H. Alford[1]

Small-scale turbulent mixing drives the upwelling of deep water masses in the abyssal ocean as part of the global overturning circulation[1]. However, the processes leading to mixing and the pathways through which this upwelling occurs remain insufficiently understood. Recent observational and theoretical work[2–5] has suggested that deep-water upwelling may occur along the ocean's sloping seafloor; however, evidence has, so far, been indirect. Here we show vigorous near-bottom upwelling across isopycnals at a rate of the order of 100 metres per day, coupled with adiabatic exchange of near-boundary and interior fluid. These observations were made using a dye released close to the seafloor within a sloping submarine canyon, and they provide direct evidence of strong, bottom-focused diapycnal upwelling in the deep ocean. This supports previous suggestions that mixing at topographic features, such as canyons, leads to globally significant upwelling[3,6–8]. The upwelling rates observed were approximately 10,000 times higher than the global average value required for approximately $30 \times 10^6 \, \text{m}^3 \, \text{s}^{-1}$ of net upwelling globally[9].

The oceanic overturning circulation redistributes heat, carbon and nutrients throughout the oceans, which is critical in regulating the global climate. This circulation involves transport across density surfaces (isopycnals) by turbulent mixing, primarily driven by breaking internal waves in waters deeper than about 2,000 m (refs. 5,9–11). These waves generate sufficient energy to upwell abyssal waters across the deep stratification to the ocean's ventilated upper layers[12]. However, measurements have shown that turbulent diffusivity in the ocean interior is significantly smaller than that needed for the buoyancy budget, suggesting that regions of intense mixing exist to maintain circulation[1]. Mixing owing to breaking internal waves typically intensifies towards the seafloor, particularly around rough topographic features[13–16]. A one-dimensional mixing profile integrating these observations in the ocean interior implies a divergent turbulent buoyancy flux leading to downwelling[2,3]. Thus, internal wave breaking in the interior might not produce the required upwelling as previously thought, leaving open the question of how deep waters return to the surface.

It has been hypothesized that upwelling may be confined to near-bottom regions within which the buoyancy flux perpendicular to the boundary must decrease to zero owing to the requirement of no flux through the boundary itself[4,5,17]. The convergent turbulent buoyancy flux leads to upwelling near the boundary. Although the no-flux condition assumes no geothermal heat flux through the seafloor, inclusion of a geothermal heat flux reinforces upwelling at the boundary by warming deep waters from below[5,18].

Microstructure measurements[3,7,19,20] and numerical simulations of passive-tracer releases[21,22] have suggested that boundaries may have an important role in upwelling. Observations[6,15] and subsequent numerical simulations of near-boundary flow[23] have suggested a vigorous exchange of mixed fluid with the interior, pointing to the importance of adiabatic processes. Together, these results reinvigorate fundamental debates on the nature of boundary mixing[24,25]. However, all evidence has, so far, been inferred indirectly and based on a range of debatable assumptions.

The near-bottom upwelling region needs to be reconciled with previous definitions of the boundary layer. A frictional bottom boundary layer owing to a uniform shear flow above a flat surface is well mixed and homogeneous. In the context of mixing driven by geostrophic flows[26], the boundary layer can be defined by the Ekman layer. One-dimensional models of steady flow on a planar slope give rise to a weakly stratified near-boundary region labelled as the bottom boundary layer[4,17,27]. These well-mixed regions are rarely observed over sloping bottoms, so recent work has instead referred to the bottom boundary layer as the region where the diapycnal transport is upwards[28]. However, above sloping topography where three-dimensional processes such as re-stratification, internal wave breaking and boundary–interior exchange have a role, the physics that gives rise to a convergent buoyancy flux near the bottom becomes murky. Because of the lack of a clear definition of this near-bottom region and because it is not well mixed owing to the rapid exchange with the interior, we refrain from referring to it as a bottom boundary layer.

[1]Scripps Institution of Oceanography, University of California San Diego, La Jolla, CA, USA. [2]Department of Earth System Science, University of California Irvine, Irvine, CA, USA. [3]Department of Earth, Atmospheric and Planetary Sciences, Massachusetts Institute of Technology, Cambridge, MA, USA. [4]Laboratoire d'Océanographie Physique et Spatiale, Univeristy of Brest, CNRS, Ifremer Centre de Bretagne, Plouzané, France. [5]Department of Geography, University of Exeter, Exeter, UK. [6]Department of Earth and Environment, Boston University, Boston, MA, USA. [7]National Oceanography Centre, Southampton, UK. [8]Royal Netherlands Institute for Sea Research (NIOZ), Den Burg, the Netherlands. [9]Department of Physical Oceanography, Woods Hole Oceanographic Institution, Woods Hole, MA, USA. [10]Ocean and Earth Science, University of Southampton, Southampton, UK. ✉e-mail: bwynneca@ucsd.edu

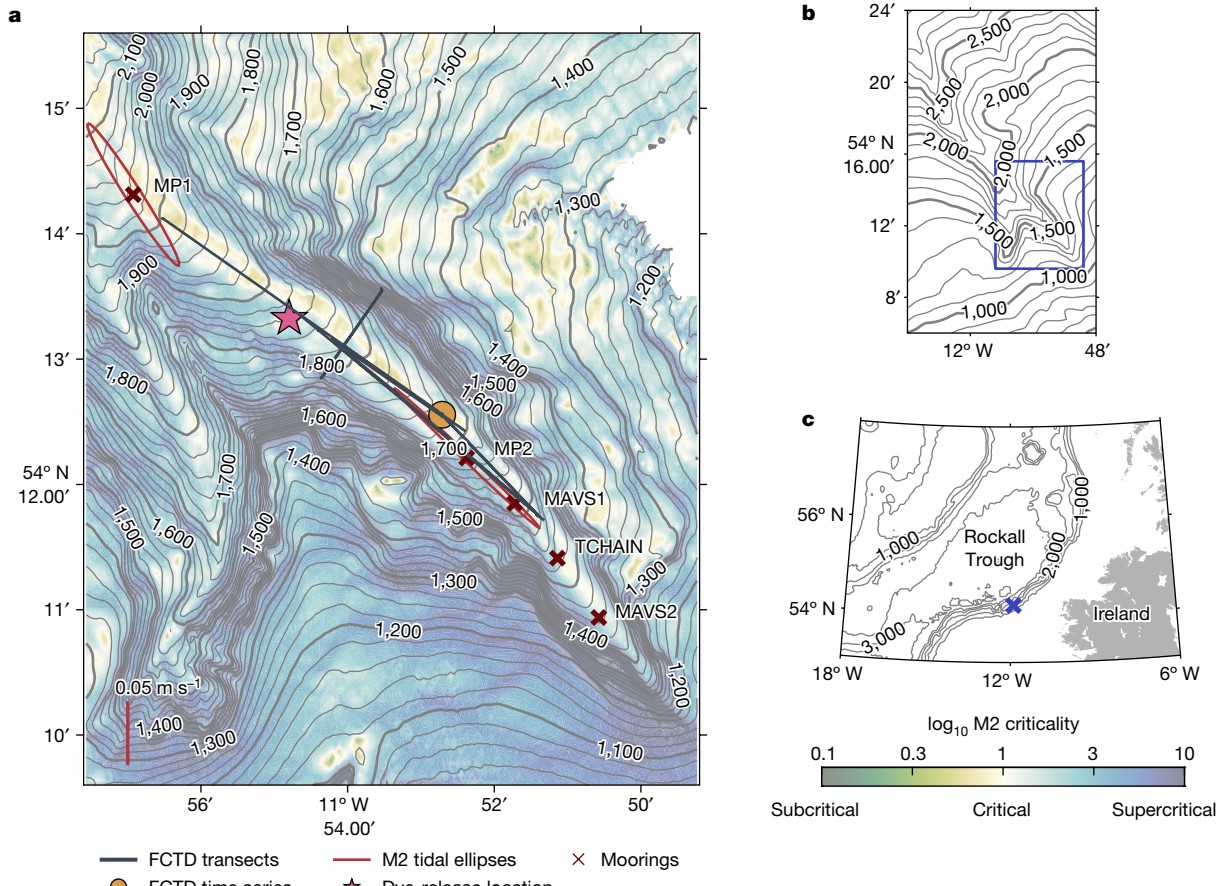

**a**, Canyon bathymetry in the vicinity of the measurements used for this study was measured using a shipboard multi-beam during the BLT Recipes experiment. Thin contours are every 20 m and thick contours are every 100 m. The locations of the FCTD transects (black lines), 12-hour time-series station (orange circle), moorings (MP1, MP2, MAVS1, TCHAIN and MAVS2) and dye release are shown. M2 tidal ellipses for the MP1 and MP2 moorings estimated from the average velocity within 100 m of the seafloor are shown

**Fig. 1 | Overview of the study region, locations of moorings and dye survey observations.** with a scale bar of 0.05 m s⁻¹. The shading shows the logarithm of slope criticality to the M2 internal tide given the time-mean stratification from shipboard CTD stations during the 5-week-long experiment (not shown). **b**, Bathymetry of the full canyon based on global bathymetry products[49,50]. Thin contours are every 100 m and thick contours are every 500 m. The blue box marks the region shown in **a**. **c**, Bathymetry of the Rockall Trough based on global bathymetry products[49,50], with contours every 1,000 m. The land is coloured in grey. The location of the canyon is marked by the blue cross.

## Experiment and study region

This study presents results from a dye release conducted as part of the Boundary Layer Turbulence and Abyssal Recipes (BLT Recipes) experiment. A primary goal of the experiment was to provide observational confirmation of vigorous near-boundary abyssal upwelling and to explain the physics driving it. The experiment site chosen was the Rockall Trough (Fig. 1), given its relatively flat, deep interior and steep sides. These characteristics enable a clear distinction between interior and boundary processes, which is essential for determining the driving forces behind the transformation of deep water. In particular, the study was conducted in a narrow, slope-incising canyon on the eastern side of the Rockall Trough. The canyon is 9-km wide at the mouth and 32-km long from the 2,900-m isobath to the tip of the eastern branch at the 1,200 m isobath. On average, the canyon walls rise 400 m above the thalweg. The canyon splits into a southwards and southeastwards branch around 54° 14.0′ N, 11° 57.1′ W. Our measurements focused on the southeastwards branch of the canyon.

The BLT canyon, like many other canyons[6,29,30], shows vigorous along-canyon flow that oscillates between the up- and down-canyon directions with twice-a-day and once-a-day frequencies (Fig. 2 and Extended Data Fig. 1d) associated with internal tides. Internal tides are generated when the depth-independent (barotropic) tide flows over seafloor features. The observed kinetic energy is two to three times

larger than the barotropic tide, and rises and falls according to the barotropic spring–neap cycle (Extended Data Fig. 2), indicating that flow within the canyon is owing to internal tides that are generated locally as opposed to propagating in from elsewhere. These motions strongly advect the temperature structure both laterally and vertically. The vertical displacements of 200 m are apparent in Extended Data Fig. 1. Because the up-canyon flow is sheared, differential advection reduces stratification (Extended Data Fig. 1b). Eventually, this causes warm water to be swept below cold water each cycle following peak up-canyon flow (Extended Data Fig. 1, zoom panel), causing strong turbulence (Extended Data Fig. 1b). This mechanism, known as convective instability, is demonstrated more quantitatively to be acting as the internal tide-breaking mechanism in our canyon (A.C.N.G. et al., manuscript under review) and has been seen atop numerous other continental slopes[29,31,32].

The tendency for internal tides to break when interacting with topography depends on the ratio of the topographic slope ($\alpha$) to the propagation angle

$$s = \sqrt{\frac{\omega^2 - f^2}{N^2 - \omega^2}} \tag{1}$$

of the internal wave ($s$, the criticality), where $\omega$ is the internal wave frequency, $f$ is the local inertial frequency and $N$ is the buoyancy frequency.

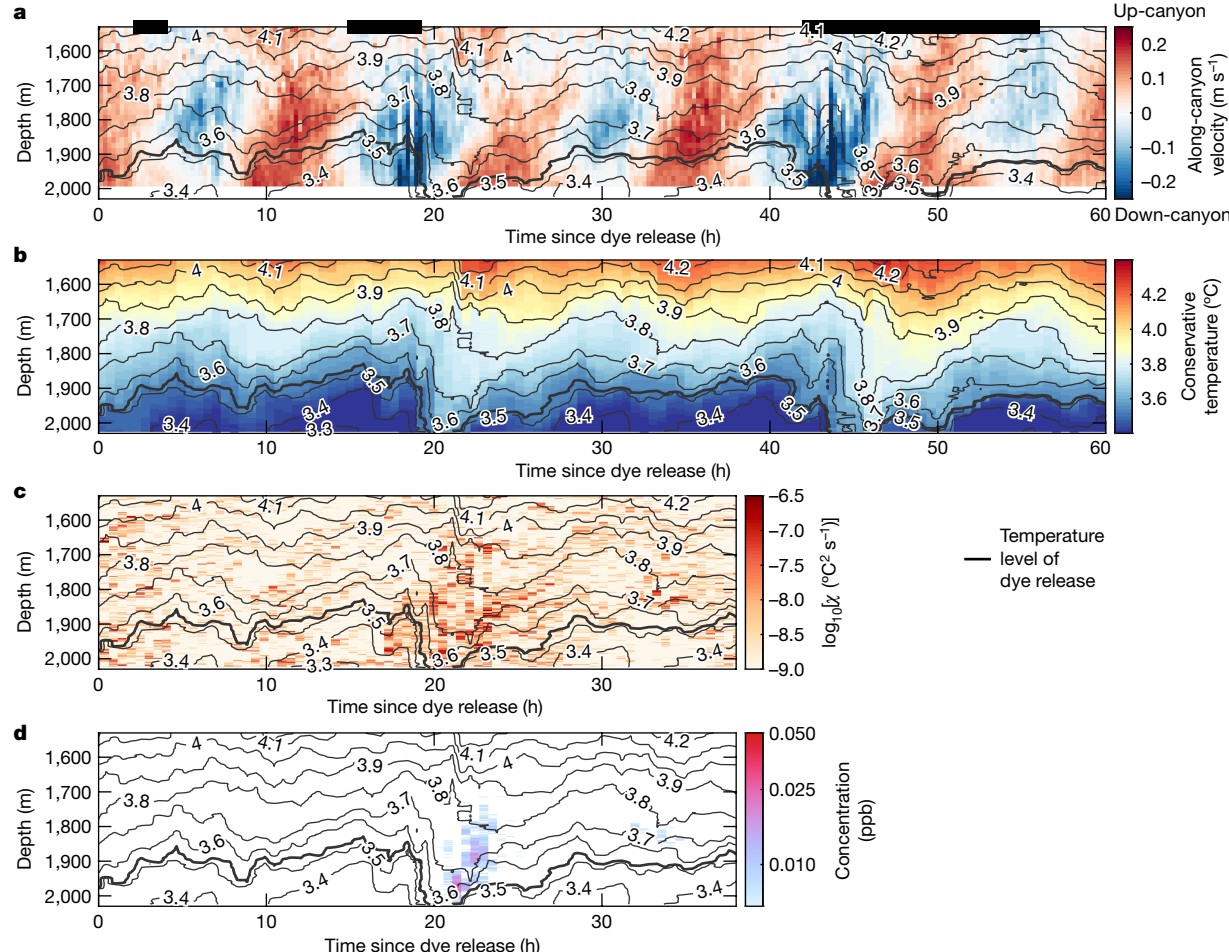

**Fig. 2 | Moored measurements of velocity, temperature, turbulent dissipation and dye concentration from the MP1 moored profiler mooring.**
**a**–**d**, Along-canyon velocity (up-canyon is positive; **a**), conservative temperature (**b**), dissipation rate of temperature variance on a logarithmic scale (**c**) and dye concentration on a logarithmic scale in ppb (**d**) from MP1. The threshold for dye concentration detection for the fluorometer onboard the MP1 is 0.006 ppb.

Thin black contours show every 0.1 °C. The release isotherm of 3.53 °C in conservative temperature is shown with the thick black contour. The thick black lines along the top of **a** indicate times of FCTD transects shown in Fig. 3. The battery for dissipation and fluorometer measurements stopped around 38 hours after the dye release.

To estimate criticality, we use the time-mean buoyancy frequency estimated from shipboard conductivity, temperature and depth (CTD) casts collected within and around the canyon during the 5 weeks of the BLT Recipes experiment. The canyon thalweg, with an average slope of 4°, has a similar slope to the internal semi-diurnal (M2) tides ($\alpha/s \approx 1$), implying local generation and critical reflection of the internal tide within the canyon. The canyon side walls are supercritical ($\alpha/s > 1$; blue shading in Fig. 1), which tends to lead to reflection and scattering of incident waves.

At longer than tidal timescales, the time-mean flow is up-canyon (velocity profiles in Fig. 4 and time series in Extended Data Fig. 3). Low-pass filtered isotherm displacements (Extended Data Fig. 4) correspond to adiabatic velocities (movement of isopycnals without mixing) smaller than the vertical component of the up-canyon flow (Extended Data Fig. 3), suggesting that the dominant balance within the canyon is between the in-flow and diapycnal flow (flow across isopycnals; Methods). In this regard, the system resembles a miniature version of the Brazil Basin system[3,8]. We measured the diapycnal flow directly with the dye release described below.

## Diapycnal upwelling measured by dye

In the BLT Recipes experiment, fluorescein dye was released near the seafloor in the centre of the canyon to measure the change in density of

water parcels and, in turn, the diapycnal upwelling velocity. This experiment differs from previous dye releases as it was conducted in deep water rather than in the coastal ocean[33–35]. Previously, deep-water-mass transformation has been observed using long-term chemical tracer experiments[2,36] where the tracer was released at least 500 m from the seafloor. Measuring chemical tracers requires water samples, the collection of which is an inherently slow process with low spatial resolution. A fluorescent dye, however, can be measured continuously with a fluorometer, allowing for significantly higher spatial and temporal resolution and facilitating direct observations of the effect of turbulent mixing over short timescales. Dye becomes undetectable much faster than a chemical tracer owing to the dye's much higher detection threshold (about $10^{-3}$ parts per billion (ppb) versus about $10^{-9}$ ppb), such that the duration of the experiment is on the order of days rather than months. Chemical tracers average together the impacts of turbulence over much longer timescales and greater distances from the bottom than a dye. The rapid changing of the shape and size of the dye patch makes estimating the diapycnal diffusivity experienced by the dye more difficult than for a long-term tracer. Instead, we can observe the diapycnal transformation of the dye close to the topography, averaged over the duration of the experiment, which may include upwelling and downwelling components depending on its extent.

The dye was released in 1,870 m of water, approximately 10 m above the canyon floor and at a conservative temperature of 3.53 °C. The rapid

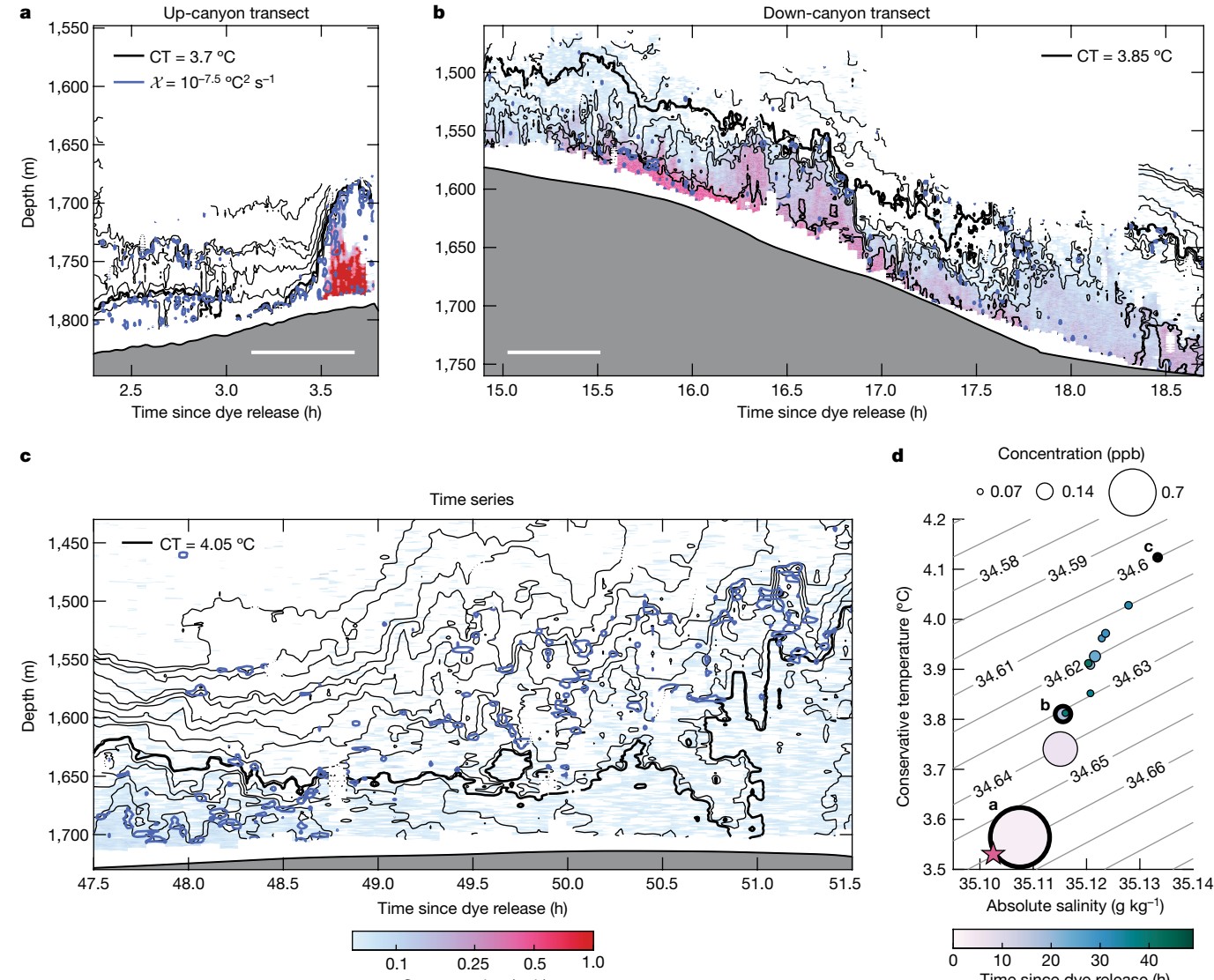

**Fig. 3 | Observations of dye concentration demonstrating upwelling of dye in density space. a**–**c**, Dye concentration as measured during transects at 2.5 hours (**a**), 15 hours (**b**) and 47.5 hours (**c**) after release. Dye concentration is shown on a logarithmic scale in ppb in colour, isotherms whose mean depths are separated by 10 m are shown with thin black contours and the $\chi = 1 \times 10^{-7.5} \, {}^\circ C^2 \, s^{-1}$ contour is shown in blue on each panel denoting regions of enhanced turbulent mixing. The thick black contours indicate reference conservative temperatures (CT) 3.7 °C, 3.85 °C and 4.05 °C for **a**–**c**, respectively. The threshold of dye concentration detection for the fluorometer onboard the FCTD is 0.06 ppb. Scale bars in **a** and **b** are 0.5 km and denote an estimate of the horizontal length of the transects. **d**, Temperature–salinity diagram of the centre of mass for each transect. Scatter points are coloured by time since the dye release in hours and the size of the data point indicates the average concentration during the transect. Scatters with thick outlines are labelled a, b and c, and correspond to the examples shown in **a**–**c**, respectively. The temperature and salinity at which the dye was released is marked with the pink star. The grey contours show lines of constant potential density anomaly in kg m$^{-3}$. See Extended Data Fig. 5 for all transects and Extended Data Fig. 6 for the full temperature–salinity distribution of each transect.

profiling 'fastCTD' (FCTD; details in Methods) carried a fluorometer, enabling high-spatial-and-temporal-resolution profiles of dye down to 2,200-m depth. The FCTD also carried a micro-conductivity probe that measured the dissipation rate of temperature variance ($\chi$), a measure of turbulent mixing. We focus on the transformation of the dye within the first 3 days after release.

To quantify the rate of flow across isopycnals, we track the evolution of the dye in density space using the dye-weighted average (Methods). We are interested here in the diapycnal transformation of the dye, and not how the dye moves up and down due to the tides in physical space. We use the density variable of potential density anomaly, $\sigma_\theta = \rho_\theta - 1{,}000$ kg m$^{-3}$, where $\rho_\theta$ is the potential density and $\theta$ is the potential temperature, both referenced to a pressure of 1,500 db.

Although seawater density depends on both temperature and salinity, density is approximately linear in temperature in this region[37], permitting us to use temperature and density interchangeably to characterize upwelling.

The first moment of the potential density anomaly, $\overline{\sigma_\theta}$, estimated using the tracer-weighted average, represents the centre of mass of the dye in density space. By estimating how the density of dyed waters decreased with time, we can infer an upwelling rate across isopycnals (Extended Data Fig. 7). The first observation of the dye was made 2 hours after release (Fig. 3a). The dye cloud was still very concentrated at this time. The centre of mass was at a potential density anomaly of $\overline{\sigma_\theta} = 34.660$ kg m$^{-3}$. Little mixing with lighter waters had occurred since the dye release at $\overline{\sigma_\theta} = 34.661$ kg m$^{-3}$ (Fig. 3d). The isotherms

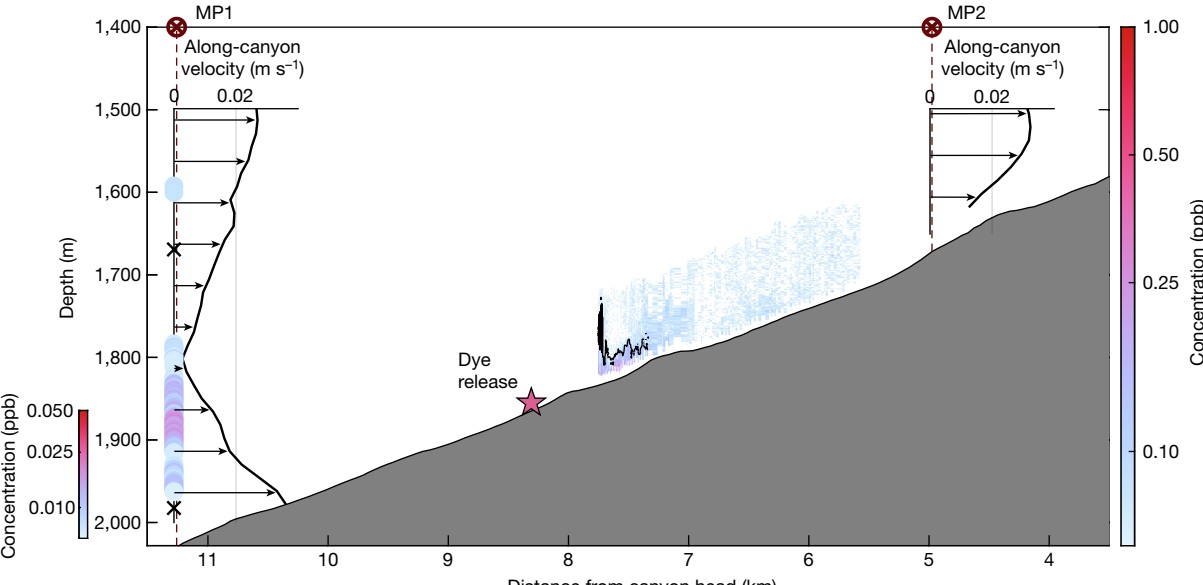

**Fig. 4 | Along-canyon overview of the observations showing the spread of dye along the canyon.** The inset axes show time-mean profiles of along-canyon velocity at MP1 (located at 11.3 km along the canyon, averaged over the full 1-week record) and MP2 (located at 4.8 km along the canyon, averaged over the full 3-month record; MP2 was deployed after MP1 was taken out of the water). Dye concentration from an example transect 20 hours after the dye release along the canyon axis is shown between 5.5 km and 8 km, with the highest concentrations shown with dark pink colours (see Extended Data Fig. 5d for

more detail). The 3.85 °C isotherm is contoured in black. The coloured dots show the dye concentration measured using the fluorometer on the MP1 mooring also at 20 hours after release. Colour scales for the dye transect and moored profile dye concentrations are different and the detection threshold is 0.06 ppb for the FCTD measurements and 0.006 ppb for the MP1 measurement. Depths of the 3.85 °C and 3.53 °C isotherms 20 hours after release are marked with black crosses on the MP1 profile.

were stretched vertically, indicative of a large overturning event due to internal wave breaking on the slope (as seen in the moored time series (Extended Data Fig. 1)). Vigorous turbulent mixing was seen along the upper edge of the bore, over 100 m above the seafloor (Fig. 3a, blue contour). The dye probably mixed strongly across this interface. Fifteen hours after release, the dye was observed as the ship travelled down the canyon (Fig. 3b). The dye had become lighter since the up-canyon transect, and the centre of mass was at $\overline{\sigma_\theta} = 34.632$ kg m$^{-3}$. Over the subsequent observations, the dye became more diffuse and spread vertically and horizontally, as seen during the FCTD time series completed at the end of the survey (Fig. 3c).

Using the rate of change of the observed dye-weighted density and the averaged vertical density gradient, the diapycnal upwelling rate of the dye ($w_{dye}$) was estimated using four different methods with values ranging from 51 m per day (m d$^{-1}$) to 325 m d$^{-1}$ (Table 1 and Methods). Despite the large spread, all the methods indicate significant upwelling of $\mathcal{O}$(100 m d$^{-1}$) near the bottom over the 3 days of the experiment.

Velocity measurements from both the moored profiler (MP) moorings, MP1 (a 1-week record) and MP2 (a 3-month record), show that the record-mean along-canyon flow was up-canyon within 500 m of the seafloor and the density decreased towards the head of the canyon (Fig. 4). The momentum balance driving the up-canyon flow is currently under investigation. Candidate driving forces include (1) large-scale pressure gradients set by the mesoscale and/or mean circulation fields and (2) convergent momentum stresses[38]. Regardless, as described above and in Methods, to maintain this along-canyon flow down the density gradient on longer than tidal timescales (Extended Data Fig. 3), there must be mixing across isopycnals otherwise adiabatic upwelling would cause isotherms to rise over long timescales, which is not observed[7,26,39] (Extended Data Fig. 4). The along-canyon velocities from the week-long MP1 record during the experiment are equivalent to vertical upwelling velocities of 50–100 m d$^{-1}$, consistent with the dye result (Extended Data Fig. 3). By contrast, estimates of the vertical adiabatic flow (which, if persistently upwards, could prevent

the need for diapycnal mixing as noted above) are significantly smaller, with both positive and negative values. Although these estimates were not exactly at the same time and location as the dye release, they do strongly suggest that the observed dye warming is interpretable as a diapycnal velocity (Extended Data Fig. 3 and Methods).

Assuming the turbulent buoyancy flux to be bottom enhanced above a sloping bottom boundary, theories predict downwelling atop a thin upwelling region[4], as discussed above. Our dye results reflect the time- and depth-integrated effects of advection and diffusion on the initial patch; hence, the rate of change of the dye-weighted density can be related to the average effects of all upwelling and downwelling processes occurring within the extent of the dye patch from release to sampling[40]. Given the patch's location within $\mathcal{O}$(100 m) from the seafloor during the short period over which we could observe it, our finding of strong upwelling is consistent with the theoretical predictions[22,41]. It is conceivable that our estimate of increased dye-weighted density (Extended Data Fig. 7) at later times could be indicative of downwelling as the patch occupies more volume above the region of convective instability where upwelling is seen. However, our error bars are too great to be sure. The dye-based estimate of the upwelling averaged over all transects is then a lower bound on the magnitude of the local near-bottom upwelling rate.

## Exchange with the interior

We observed a patch of the dye extending away from the boundary into the ocean interior, consistent with past inferences[6] and numerical modelling[23]. The observation is significant because it demonstrates rapid fluid exchange with the interior as the mechanism for re-stratifying the near-bottom region and provides direct evidence that one-dimensional treatments of the boundary are insufficient.

During the dye experiment, the MP1 mooring was located 3 km down-canyon of the release site. The dye reached the mooring twice, once 20 hours after release and once 32 hours after release (Fig. 2).

**Table 1 | Estimates of diapycnal upwelling rate from the dye**

| Method | $w_{dye}$ (m d$^{-1}$) |
|---|---|
| Weighted linear regression | 250±75 |
| Unweighted linear regression | 101±50 |
| Average of pairwise estimates | 125±31 |
| Change in height above bottom | 64 |

The results of the four different estimates of the diapycnal upwelling rate ($w_{dye}$); a linear regression of the centre of mass weighted by the standard deviation of the dye-weighted average, an unweighted linear regression to the centre of mass, an average over all estimates of the diapycnal velocity between each pair of transects, and the change in depth of the centre of mass between the first and last observations of the dye.

Importantly, the dye was detached from the seafloor, forming a tongue of dye extending into the interior. The FCTD survey of the dye 20 hours after release (Fig. 4) captured the up-canyon edge of the dye. Dye concentrations were much lower at the mooring than at the FCTD location, suggesting that only the edge of the patch reached the mooring location. These two sets of measurements indicate that the patch size 20 hours after release was at least 4 km in length (the fluorometer on the mooring was 10 times more sensitive than the one on the FCTD (Methods); therefore, additional weakly concentrated dye further up the canyon may have gone undetected by the FCTD). In the FCTD and mooring records, the dye was colder than the 3.85 °C isotherm. In the mooring record, the dye remained warmer than the release isotherm, suggesting that it had not downwelled since its initial release. This direct evidence of the export of mixed fluid from the boundary underscores the need for a four-dimensional understanding of the tidal processes responsible for turbulence. This view differs strongly from the one-dimensional paradigm[42] and supports previous observations[6,15] and subsequent numerical simulations of near-boundary flow[23] that suggest a vigorous exchange of mixed fluid with the interior, possibly driven by a convergent near-boundary flow[6].

## Summary

This work provides direct evidence of diapycnal upwelling along a sloping canyon in the deep ocean. Upwelling velocities of $\mathcal{O}(100 \text{ m d}^{-1})$ show that processes at steeply sloping topography led to rapid upwelling of deep water to lighter density classes. Diapycnal upwelling was coupled with adiabatic advection that transported dye away from the boundary along isopycnals. Although observational limitations in this study lead to uncertainty in the dye-inferred upwelling experienced by the dye, the similarity between independent estimates of dye-based upwelling and diapycnal upwelling inferred from the measured along-canyon flow is encouraging.

Previous direct estimates of water-mass transformation in the deep ocean have come from chemical tracer releases that focused on interior transformation and have much larger spatial and temporal averaging scales than this dye release and average over regions of near-boundary upwelling and above-boundary downwelling, as opposed to our more focused observations that more strongly weight near-boundary upwelling[2,36]. These chemical tracer releases measured net downwelling, although some tracer may have moved upwards along the seafloor[2].

Previous estimates of upwelling near the bottom were smaller in magnitude but were observed over larger spatial and temporal scales than our experiment. The estimated average upwelling value required to maintain global deep-ocean density stratification is $1 \times 10^{-2}$ m d$^{-1}$ (ref. 9). In a fracture-zone canyon of the Mid-Atlantic Ridge, the magnitude of the up-canyon velocity was similar to that seen here. However, the shallower bathymetric slope led to a relatively smaller upwelling velocity of 1.7 m d$^{-1}$ (ref. 7). Diapycnal upwelling in the Mid-Atlantic Ridge canyon is controlled by a decrease in the volume available for

mixing as the canyon narrows (hypsometry)[8]. In Monterey Canyon, estimates of the net upwelling ranged from 0.2 m d$^{-1}$ to 1 m d$^{-1}$ and were also driven by the hypsometry[6].

Our observations provide a direct estimate of the upwelling at a bottom boundary. However, more observational experiments in various physical environments are necessary to estimate the near-boundary's contribution to net global upwelling accurately.

Our results support previous theoretical studies that propose that near-boundary mixing is important for deep-ocean upwelling[4,5,17,43]. Specifically, our results provide direct evidence in support of these studies' key prediction: to overcome the substantial interior ocean downwelling implied by bottom-enhanced mixing, the net global upwelling of about $30 \times 10^6$ m$^3$s$^{-1}$ requires the existence of much more rapid upwelling near the sloping seafloor. An important limitation of these previous studies is that they focused on large-scale water-mass transformations due to parameterized mixing, primarily due to tides; by contrast, here, we directly observe these tidally modulated water-mass transformation processes.

Complementary observations from the FCTD and long-term moorings during our experiment found that periods of intense mixing and upwelling were associated with convective breaking of the internal tide within the canyon[37] (A.C.N.G. et al., manuscript in review). Similar convective mixing has been observed on sloping boundaries elsewhere, within canyons[29], on continental shelves and slopes[31,32,44,45], and in lakes[46], suggesting that this process may be of broad importance.

Our work suggests that acknowledgement of the time-variable, three-dimensional nature of near-boundary mixing processes may be essential to adequately representing and understanding near-bottom upwelling physics. On the basis of our observations and the widespread distribution of submarine canyons across the globe[47], previous global scalings based on weaker upwelling velocities may be underestimates[6,7,48]. Upwelling within these canyons may have a more significant role in overturning deep water than previously thought.

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

# Methods

## BLT Recipes experiment

The data presented here were collected during expedition DY132 of RRS *Discovery*[51] between 19 June 2021 and 29 July 2021, supported by the Natural Environment Research Council (NERC) and National Science Foundation (NSF)-funded Boundary Layer Turbulence and Abyssal Recipes project. The experiment consisted of a dye release together with surveys and moorings measuring hydrographic data, velocity, and shear and temperature microstructure.

**Dye release.** A fluorescent dye (fluorescein $C_{20}H_{10}O_5Na_2$) was used for this experiment. The mixture used for the release consisted of 64 l of isopropyl, 6 l of seawater and 149 l of 40% fluorescein liquid. The mixture was chosen to be denser than the local water at the desired depth to ensure that it was released properly and did not rise without mixing. The dye-release system (InkBot; University of Exeter) comprises a 219-l drum equipped with a Seabird SBE 911plus CTD, an AQUAtracka III fluorometer (Chelsea Technologies Group), an altimeter and an OCEANO 2500S Universal acoustic release. For deployment, the InkBot was attached to the CTD sea cable and lowered to 10 m above the bottom using the ship winch. Once in position, the drum was emptied by simultaneously flipping the drum and opening the drum lid using the acoustic release. The drum was upside down for 15 min, while readings from the onboard CTD enabled the winch operator to ensure that the InkBot remained at the release temperature. The dye release began at 03:01 am on 1 July 2021. The BLT dye release is novel in its depth and deep-ocean environment, where common dye sampling techniques such as aerial multi-spectral imagery could not be used. Instead, we performed rapid-repeat fluorometer measurements using the FCTD and operated the ship in a pattern that dynamically followed the dye.

**FCTD.** The FCTD system[52] is a tethered profiler with a Seabird SBE49 CTD, a dual needle micro-conductivity probe (built in-house at Scripps Institution of Oceanography by the Multiscale Ocean Dynamics group) and, for this experiment, a Turner C-FLUOR fluorometer and an altimeter. The instrument was raised and lowered with a direct drive electric winch at vertical speeds of approximately 3 m s$^{-1}$ while the ship was steaming at a speed of 0.5–1 knots. Data from both up- and down-casts were used. The goal of the FCTD survey was to sample the dye release as many times as possible before dye concentrations became too low to be detected. Transects were focused along the canyon axis to transit from one end of the dye patch to the other before turning around and repeating. In addition, two cross-canyon transects and a 12-hour time-series station were completed to better understand the full extent of the dye patch. During a transect, the FCTD profiled over a roughly 100-m vertical range and to within less than 10 m of the seafloor. Data from the FCTD were monitored and used interactively to guide the survey.

The micro-conductivity probe was used to estimate the dissipation rate of temperature variance ($\chi$). Conductivity gradients can be used as temperature dominates the conductivity variance[53].

**MP1 and MP2 moorings.** The MP1 mooring was deployed on the canyon axis at 2,028-m water depth at 54° 14.312′ N, 11° 56.923′ W. The mooring was 600-m tall and consisted of a downwards-looking RDI Longranger 75-kHz acoustic Doppler current profiler (ADCP) mounted in the top float and a McLane moored profiler outfitted with a Seabird SBE52 CTD, a Falmouth Scientific acoustic current meter and an epsilometer turbulence package (Multiscale Ocean Dynamics group, Scripps Institution of Oceanography)[54] to measure $\chi$ from temperature gradients, and a Turner C-FLUOR fluorometer. The profiler was deployed on 28 June 2021, and profiled continuously for 7 days, collecting one profile every 30 min. The mooring was redeployed (as MP2) on 7 July 2021, without the fluorometer and epsilometer at 54° 10.938′ N, 11° 50.572′ W at a depth of 1,676 m. This deployment lasted until 6 October 2021.

**MAVS1, TCHAIN and MAVS2 moorings.** The MAVS1 and MAVS2 moorings were deployed for approximately 3 months from 6 July 2021 to 7 October 2021. The MAVS moorings were each 300-m tall and consisted of 8 modular acoustic velocity sensors (MAVS; Woods Hole Oceanographic Institute), 80 RBR Solo or Seabird SBE56 thermistors, a Seabird SBE37 CTD and an RDI Longranger 75-kHz ADCP mounted on the top float. MAVS1 was deployed at 54° 11.849′ N, 11° 51.719′ W at 1,612-m depth and MAVS2 was deployed at 54° 10.938′ N, 11° 50.572′ W at 1,466-m depth. The TCHAIN mooring (Hans van Haren, Royal Netherlands Institute for Sea Research), deployed at 54° 11.413′ N, 11° 51.137′ W at 1,529 m, consisted of a 150-m thermistor chain with 102 pre-attached thermistors and a 75-kHz RDI ADCP mounted on the top float. TCHAIN was deployed from 6 July 2021 to 11 August 2022. Detailed analysis of the long-term moorings will be discussed elsewhere.

**Fluorometer calibration.** The Turner C-FLUOR fluorometer calibration uses the linear relationship $C = (V - a)b$ where $C$ is the concentration in ppb, $V$ is the measured voltage, $a$ is an offset and $b$ is the calibration coefficient. The offset voltage was lower than the factory-provided value for both fluorometers used. For MP1, we used profiles before the dye release, and for the FCTD we used the upper 500 m of down-casts from the surface to measure the mean background voltage when zero dye was present. We used the factory calibration coefficient for both fluorometers. This gave calibrations of $C_{FCTD} = (V_{FCTD} - 0.0140) \times 31.1831$ and $C_{MP} = (V_{MP} - 0.0139) \times 31.3309$. The minimum detectable level of the fluorometers was determined as three standard deviations above the mean background level. This gave minimum detectable concentrations of 0.06 ppb for the fluorometer on the FCTD and 0.006 ppb for the fluorometer on MP1. The factor of 10 difference between detection limits is probably owing to individual sensor differences and an electrically noisier channel on the FCTD. Observed dye concentrations at MP1 were up to three times the detection limit. At the end of the FCTD survey, the levels were down to twice the detection limit.

**Sampling uncertainty.** Owing to high flow speeds—up to 0.4 m s$^{-1}$ along the canyon—the dye was advected rapidly along the canyon, requiring the dye survey to focus on two-dimensional transects through the patch. A source of uncertainty in our results comes from the resulting under-sampling of the dye patch. The two cross-canyon transects completed during the survey (Extended Data Fig. 5h,i) show that lateral distribution of the dye varied in time or along the length of the patch. During the first cross-canyon transect, the dye was spread fairly uniformly across the canyon width; however, in the second transect, the dye was banked on the northeast side of the canyon. Thus, along-canyon transects, which focused on the canyon axis, may at times have sampled through only the edge of the patch.

With only the velocity measurements from MP1 (down-canyon of the dye for most of the experiment), it is difficult to estimate where the dye patch might have been in the cross-canyon direction. Along-canyon transects may have sampled only the front or back edge or a small section of the patch at times, particularly during later transects when the dye patch was large. On the basis of dye measurements from MP1 (Fig. 2), the entire patch was about 4-km long 20 hours after the release. Later, along-canyon transects were all shorter than this (Extended Data Fig. 8). It is possible that MP1, located in the centre of the canyon, may have missed a large concentrated patch of dye if it were banked against one wall.

To investigate the effect of under-sampling on estimates of the centre of mass, we subsampled the cross-canyon transects (Extended Data Fig. 5h,i), the longest along-canyon transects (Extended Data Fig. 5b,c) and the time series (Extended Data Fig. 5l). Subsampling involved systematically selecting ten consecutive profiles along a transect. This approach ensured that the subsamples represented a fraction of the dye patch. The centre of mass was then estimated for each subsample. The results are shown in Extended Data Fig. 9. The effect of subsampling

varies depending on the transect. The size of these standard deviations across different samples (Extended Data Fig. 9) varies from transect to transect in a similar way to the standard deviations of the dye-weighted density of each transect (bars in Extended Data Fig. 7). This is likely because the length and temperature range of the transects impact the overall standard deviation as well. As we cannot get an accurate estimate of the error due to sampling from all of the transects, we will use the standard deviation of the dye-weighted average to estimate the error.

This method may be particularly insufficient for the shorter transects. Looking at transects e, f and g (Fig. 3e–g), the temperatures measured are all warmer than 3.85 °C (green temperature contour) and, particularly in transects f and g, dye concentrations were relatively weak. However, during transect h, which was in the cross-canyon direction, there is more concentrated dye colder than 3.85 °C. This indicates a concentrated, colder part of the dye patch not measured in the three previous transects, and the true centres of mass may be colder than was measured. There is no way for us to account for this error accurately.

Another artefact of our sampling was due to the phase of the tide. For example, during the second transect (Extended Data Fig. 5b), we sampled the dye in a down-canyon direction as the dye itself was moving down-canyon, causing the patch to appear spread over a larger extent than it was. This may be another source of error in our estimates.

Typically, long-term chemical tracer release surveys are designed to provide information on the three-dimensional spread of the tracer. Observations are often objectively mapped to higher-resolution grids[55]. These maps provide an estimate of the fraction of tracer found. Doing such an inventory is difficult, given the two-dimensional nature of our sampling pattern. However, as this study focuses on the first rather than the second moment of the tracer, the results are less impacted by outliers.

## Calculations

**Estimating adiabatic versus diapycnal upwelling.** Consider the situation in our canyon, which is equivalent to that in many past buoyancy and mass budget calculations[3,56]: a volume bounded below by a sloping seafloor, on the sides by the canyon walls and the top by a neutral surface $\gamma_o$, which is a mean distance $H$ above the bottom and has an area $A_z$ out to the location of the in-flow. A lateral flow $u_{in}(t)$ is incident beneath $\gamma_o$ through a cross-sectional canyon area $A_x$. Volume conservation for an incompressible fluid requires one of two things to happen: $\gamma_o$ can rise adiabatically at a rate $w_{adia}$, which requires no mixing, or fluid can exit the volume via a divergent turbulent buoyancy flux, $J_b$, which produces a turbulent diapycnal velocity $w^* = \frac{1}{N^2}\frac{\partial J_b}{\partial z}$. Hence, at all times, the following balance must hold:

$$\int_{A_x} u_{in}(t)\mathrm{d}A_x = \int_{A_z} w_{adia}\mathrm{d}A_z + \int_{A_z} w^*\mathrm{d}A_z. \qquad (2)$$

Assuming $u_{in}$, $w_{adia}$ and $w^*$ are constant in space, and the canyon has a rectangular cross-section with constant width, equation (2) can be simplified to

$$A_x u_{in} = A_z (w_{adia} + w^*) \qquad (3)$$

where the ratio $A_x/A_z$ is approximately equal to $\tan\alpha$, where $\alpha$ is the slope of the bathymetry. On very long timescales, the observed constancy of the ocean's stratification requires $w_{adia} = 0$, giving a balance between $u_{in}$ and $w^*$.

We verify that this balance holds at both MP1 and MP2 on the approximately 3-day timescales of the dye observations. The in-flow velocity, $u_{in}$, was estimated as the along-canyon velocity below an isotherm. The 3.7 °C isotherm and 4.2 °C isotherms were chosen for MP1 and MP2, respectively, as they were on average about 100 m above the bottom and therefore within the region where near-boundary mixing occurs.

Given the displacement of these isotherms ($\eta$), the adiabatic velocity is then $w_{adia} = \mathrm{d}\eta/\mathrm{d}t$. Velocities and temperatures were low-pass filtered at a period of 48 hours with a fourth-order Butterworth filter to remove the diurnal and semi-diurnal tides.

Extended Data Fig. 3 shows that both $u_{in}$ and $w_{adia}$ vary in time but that the dominant balance is between $u_{in}$ and $w^*$. The average of $w_{adia}$ at MP2 is approximately zero, and $u_{in}$ is within one standard deviation of the mean of $w_{adia}$ just 4% of the time. These results bolster our interpretation that in-flow is balanced by diapycnal transport on subtidal timescales as in longer-term tracer releases such as the Brazil Basin. By contrast, adiabatic motions on tidal timescales are, in fact, key for effecting the observed water-mass transformation and exchange with the interior, as argued in the main text.

**Centre of mass.** For a dye or tracer of concentration $C$, the tracer-weighted average operator is defined as

$$\overline{(\cdot)} = \frac{\iiint (\cdot)\,C\mathrm{d}x\mathrm{d}y\mathrm{d}z}{\iiint C\mathrm{d}x\mathrm{d}y\mathrm{d}z}. \qquad (4)$$

Here we use the first moment of the tracer-weighted average of the potential density anomaly ($\overline{\sigma_\theta}$), which represents the centre of mass of the dye in density space, to describe the location of the dye patch[40,41]. Ideally, the integral is performed on the full extent of the three-dimensional dye patch. In practice, however, we are limited to the spatial information of the survey. The integrals were estimated as sums in the vertical and along-transect directions for each transect through the dye patch.

**Upwelling rate.** Given the first density moment, the dye-weighted diapycnal velocity is given by $w_{dye} = -\frac{\partial_t \overline{\sigma_\theta}}{|\overline{\nabla\sigma_\theta}|}$ (refs. 40,41). As mixing acts on dye gradients as well as density gradients, the dye-weighted diapycnal velocity yields twice the dye-weighted density velocity, $\partial_t \overline{\sigma_\theta} = 2\overline{\dot{\sigma}_\theta}$, where $D\sigma_\theta/Dt = \dot{\sigma}_\theta$ is the material derivative of the potential density anomaly[40]. A total of 12 transects were completed over 3 days before dye concentrations became too low to detect. Below, we describe four methods for estimating the diapycnal upwelling rate, the results of which are shown in Table 1.

The slope of a weighted linear regression is used to estimate the evolution of the dye's centre of mass over time ($\partial_t \overline{\sigma_\theta}$; Extended Data Fig. 7). Weights chosen for the regression were $\mathcal{W} = \frac{ns_i^{-2}}{\sum_{i=1}^n s_i^{-2}}$ where $s_i$ are standard deviations of dye-weighted density for each transect $i = 1, ..., n$ and $n = 12$, such that transects with large standard deviations are weighted low. Linear regression (solid blue line in Extended Data Fig. 7), with $R^2 = 0.852$, yields $\partial_t \overline{\sigma_\theta} = -0.0341 \pm 0.0038$ kg m$^{-3}$ d$^{-1}$ where the standard error of the slope of the fit gives the error. We approximate the density gradient with its vertical component ($\overline{\partial_z \sigma_\theta}$). Here the horizontal gradient of the density is approximately ten times smaller than the vertical component and, therefore, has a negligible impact on the vertical velocity. We calculate the dye-averaged vertical density gradient for each transect and then use the weighted average over all the transects. The weights are equivalent to $\mathcal{W}$ but are a function of the standard deviations of the dye-weighted density gradient. The density gradient used for the calculation of $w_{dye}$ is then $\overline{\partial_z \sigma_\theta} = -1.4 \times 10^{-4} \pm 0.3 \times 10^{-4}$ kg m$^{-4}$. For comparison, using all the FCTD data, without weighting by the concentrations, the average density gradient is $-2.0 \times 10^{-4} \pm 6.4 \times 10^{-4}$ kg m$^{-4}$. The weighted linear regression gives an upwelling velocity of $250 \pm 75$ m d$^{-1}$, assuming that errors associated with the time rate of change of the centre of mass and the centre of mass of the vertical gradient are independent. Temporal and spatial density gradients in this location are not independent owing to the strong influence of the tide on the stratification; however, quantifying this effect on the centre of mass is difficult.

The linear regression can also be done without weighting (Extended Data Fig. 7, solid red line). In this case, $\partial_t \overline{\sigma_\theta} = -0.0207 \pm 0.0055$ kg m$^{-3}$ d$^{-1}$

and $\overline{\partial_z \sigma_\theta} = -2.0 \times 10^{-4} \pm 4.7 \times 10^{-5}$ kg m$^{-4}$, yielding an upwelling velocity of $101 \pm 50$ m d$^{-1}$. For this fit, $R^2 = 0.587$.

An alternative method for estimating the upwelling rate is to calculate pairwise estimates between transects[22]. Given 12 individual estimates of the centre of mass, there are 66 estimates of the upwelling rate between pairs of transects. The time evolution of the dye-weighted density ($\partial_t \overline{\sigma_\theta}$) is estimated by finite differencing between each observation, where the observation time is chosen to be the average time for the transect. That is, $(\partial_t \overline{\sigma_\theta})_{i,j} = \frac{(\overline{\sigma_\theta})_j - (\overline{\sigma_\theta})_i}{t_j - t_i}$. The average over all pairwise estimates is $\partial_t \overline{\sigma_\theta} = -0.0164 \pm 0.0052$ kg m$^{-3}$ d$^{-1}$ where the error is the standard error over all pairwise estimates. The spatial density gradient ($|\nabla \overline{\sigma_\theta}|$) is taken to be the average of the pair of tracer-weighted vertical density gradient estimates. The overall average for the vertical density gradient is the same as for the unweighted linear regression case. Taking the average of all 66 estimates of the upwelling rate gives $125 \pm 31$ m d$^{-1}$ where the error is the standard error on the mean.

A final, fourth estimate of the upwelling rate can be calculated from the change in depth of the centre of mass during the experiment; we consider depth changes relative to the bottom to avoid spurious upwelling due to tidal aliasing. Using the dye-weighted average of the height above bottom, the height of the centre of mass during the first transect was 28 m above the bottom. Similarly, for the final observation during the time series, the average height of the centre of mass was 151 m above the bottom. Using the same time convention as the pairwise method, the time difference between observations was 46 hours, giving an upwelling rate of 64 m d$^{-1}$. The change in density between the first and last observations was $-0.0312$ kg m$^{-3}$ d$^{-1}$.

All four of these methods are imperfect: the weighted linear regression uses weights that may not be representative of the actual sampling error; neither the unweighted linear regression nor the pairwise methods account for the sampling error at all; and the change in depth estimate does not account for the diapycnal component. However, all these methods give a significant positive velocity of $\mathcal{O}(100$ m d$^{-1})$ and together provide confidence in our assertion that we observed diapycnal upwelling.

## Data availability

The FCTD data are publicly available at https://doi.org/10.17882/98178 (ref. 57). The mooring data are publicly available at https://doi.org/10.5061/dryad.v15dv424f (ref. 58). The multi-beam bathymetry data are publicly available at https://doi.org/10.17882/99872 (ref. 59). The GEBCO global bathymetry product[49,50] used for generating the regional and basin scale maps is available at https://www.gebco.net/data_and_products/gridded_bathymetry_data/.

## Code availability

The MATLAB (R2023b) code for generating the figures and results in the article is available at https://doi.org/10.17882/98178 (ref. 57).

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

**Acknowledgements** We thank the captain and crew of RRS *Discovery*, the National Marine Facilities technicians for assistance during the experiment and the engineers of the Multiscale Ocean Dynamics group for their assistance and design and fabrication of the instrumentation. We thank A. Brousseau for designing, building and testing the dye release system in collaboration with J. Ledwell, H.M., E. Hayden and M.-J.M. This work was funded by the Natural Environment Research Council (grant NE/S001433/1) and the National Science Foundation (grants OCE-1756264, OCE-1756324 and OCE-1756251).

**Author contributions** B.L.W.-C. performed analyses, produced figures and wrote the paper with feedback and suggestions from all authors at each stage of revision. M.H.A. conceived the dye study, designed the survey plan and provided guidance on analysis, interpretation and writing. M.H.A., A.C.N.G, G.V., R.F., K.P., M.-J.M. and H.M. conceived and proposed the BLT Recipes experiment. M.H.A., A.C.N.G., G.V., K.P., M.-J.M., H.M., H.v.H., C.P.S., A.L.B., N.C., B.L.W.-C., X.R., H.F.D. and R.F. contributed to the design and execution of the experiment. M.-J.M. and H.M. performed the dye release. G.V. designed and led the deployment of the moorings. N.C. and B.L.W.-C. calibrated the dye sensors.

**Competing interests** The authors declare no competing interests.

**Additional information**
**Correspondence and requests for materials** should be addressed to Bethan L. Wynne-Cattanach.

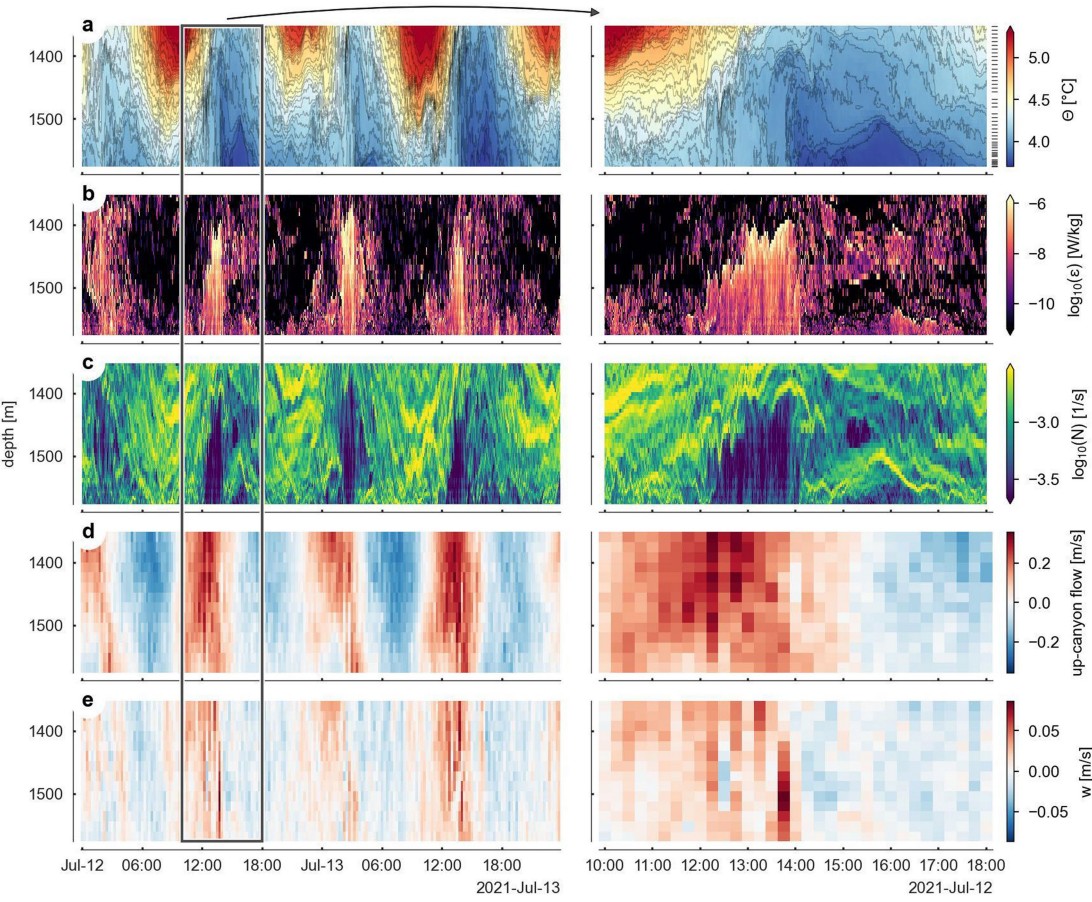

**Extended Data Fig. 1 | Two-day snapshot and 8-hour zoom of flow conditions at MAVS1.** (a) Potential temperature ($\theta$), (b) turbulent dissipation rate of kinetic energy ($\epsilon$) estimated from the sorting distance when sorting profiles into stable state, (c) buoyancy frequency $N$ calculated from the stable profile, (d) up-canyon velocity and (e) vertical velocity ($w$) at the MAVS1 mooring.

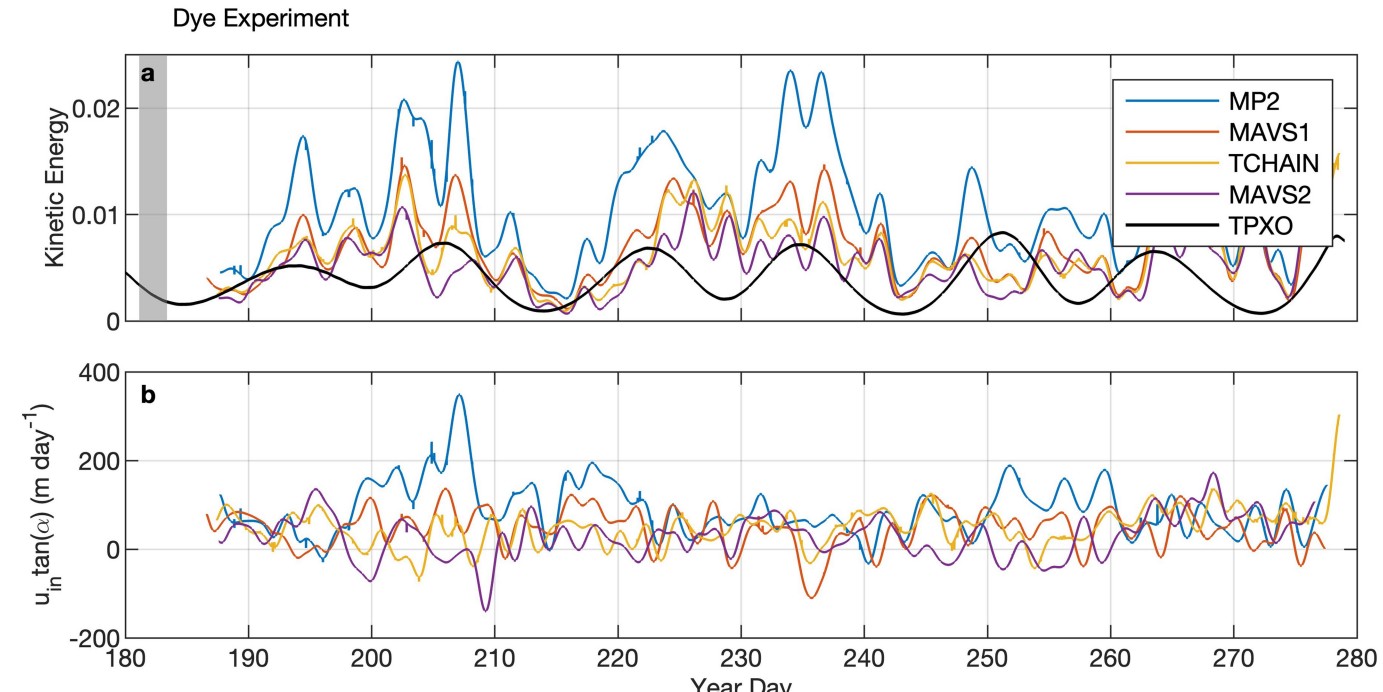

**Extended Data Fig. 2 | Kinetic energy and the in-flow velocity $u_{in}\tan\alpha$ versus time.** (a) Kinetic energy, ($\frac{1}{2}(u^2 + v^2)$ where $u$ and $v$ are the zonal and meridional velocities respectively), low-pass filtered at a period of 48 hours, averaged over the bottom 100 m from the MP2, MAVS1, TCHAIN and MAVS2 moorings (coloured lines) and the barotropic tides estimated using TPXO. The grey shaded region shows the period covered by the dye experiment. (b) Depth average of the in-flow velocity ($u_{in}\tan\alpha$) over the bottom 100 m from the four moorings, also low-pass filtered at a period of 48 hours.

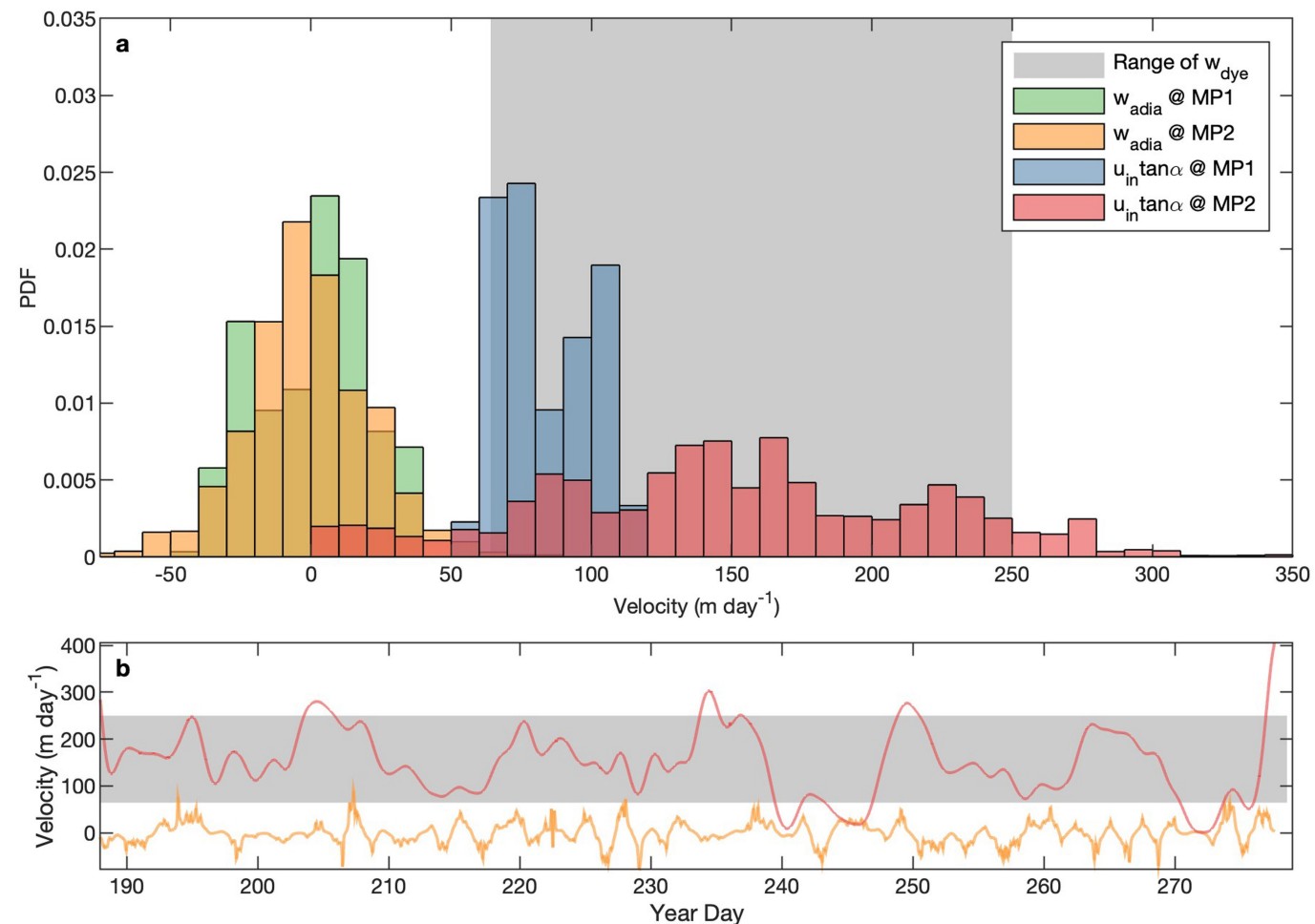

**Extended Data Fig. 3 | Probability density functions (PDF) and time series of the terms in the volume budget below a temperature surface.** (a) PDFs of the depth-averaged in-flow ($u_{in}\tan\alpha$) estimated from along-canyon velocities below 3. 7 °C at MP1 (blue) and below 4. 2 °C at MP2 (red) and the depth-averaged adiabatic vertical velocity ($w_{adia} = \frac{d\eta}{dt}$, where $\eta$ is the isotherm displacement) at MP1 (green) and MP2 (orange). The range of values of $w_{dye}$ are shown by the grey shaded region. (b) Time series of $u_{in}\tan\alpha$ (red) and $w_{adia}$ (orange) from MP2. The range of values of $w_{dye}$ are shown by the grey shaded region.

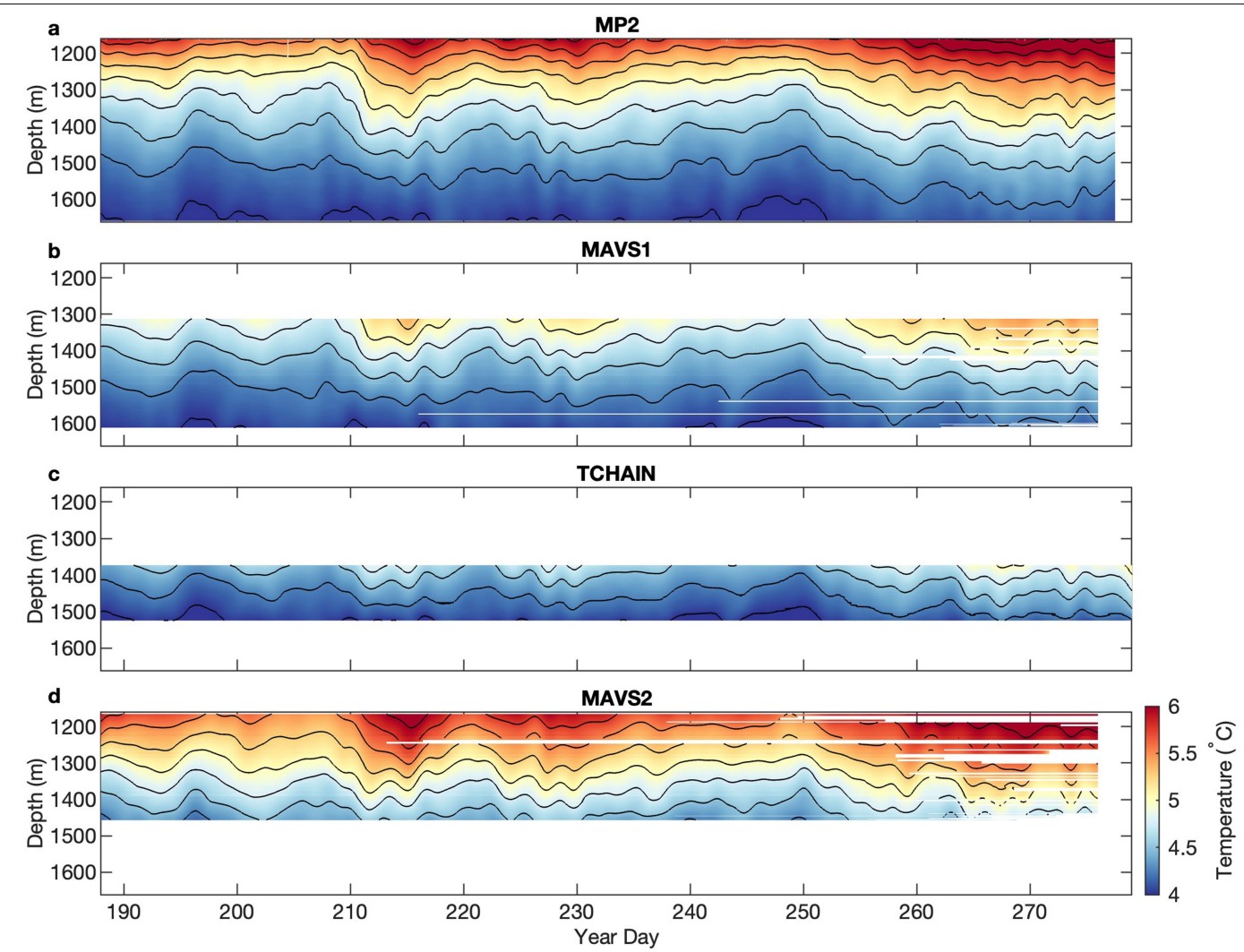

**Extended Data Fig. 4 | Temperature at each mooring, low-pass filtered at a period of 48 hours.** Temperatures from (a) MP2, (b) MAVS1, (c) TCHAIN and (d) MAVS2 low-pass filtered at a period of 48 hours. Black contours denote isotherms every 0.25 °C.

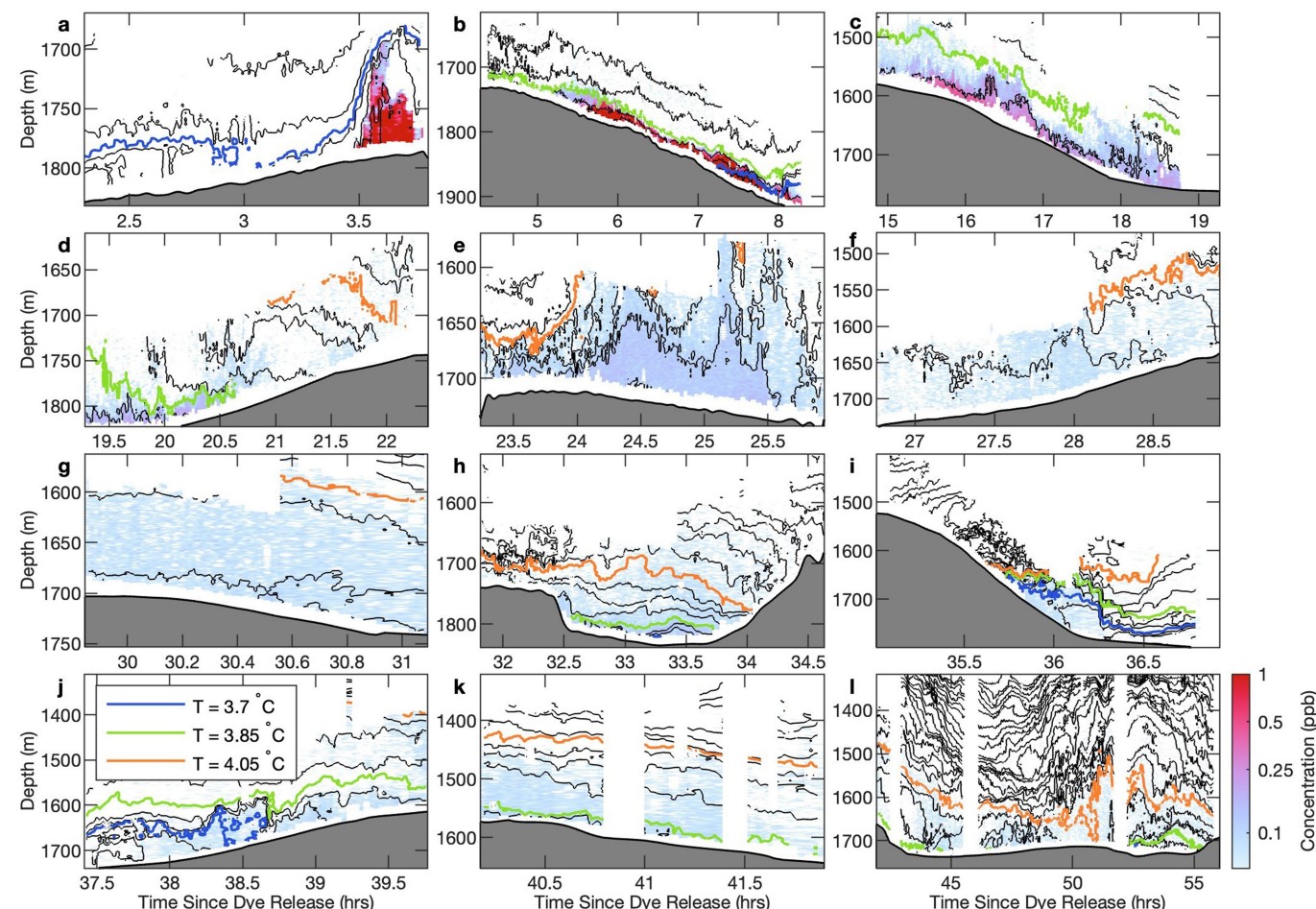

**Extended Data Fig. 5 | All transects from the FCTD survey of the dye.**
(a)-(l) Dye concentration is shown on a logarithmic scale in parts per billion (ppb) in colour, isotherms every 0.05 °C are shown with black contours, and 3.7 °C, 3.85 °C and 4.05 °C isotherms are contoured in blue, green and orange respectively. Note the different aspect ratios in each panel. Bathymetry is shown in grey.

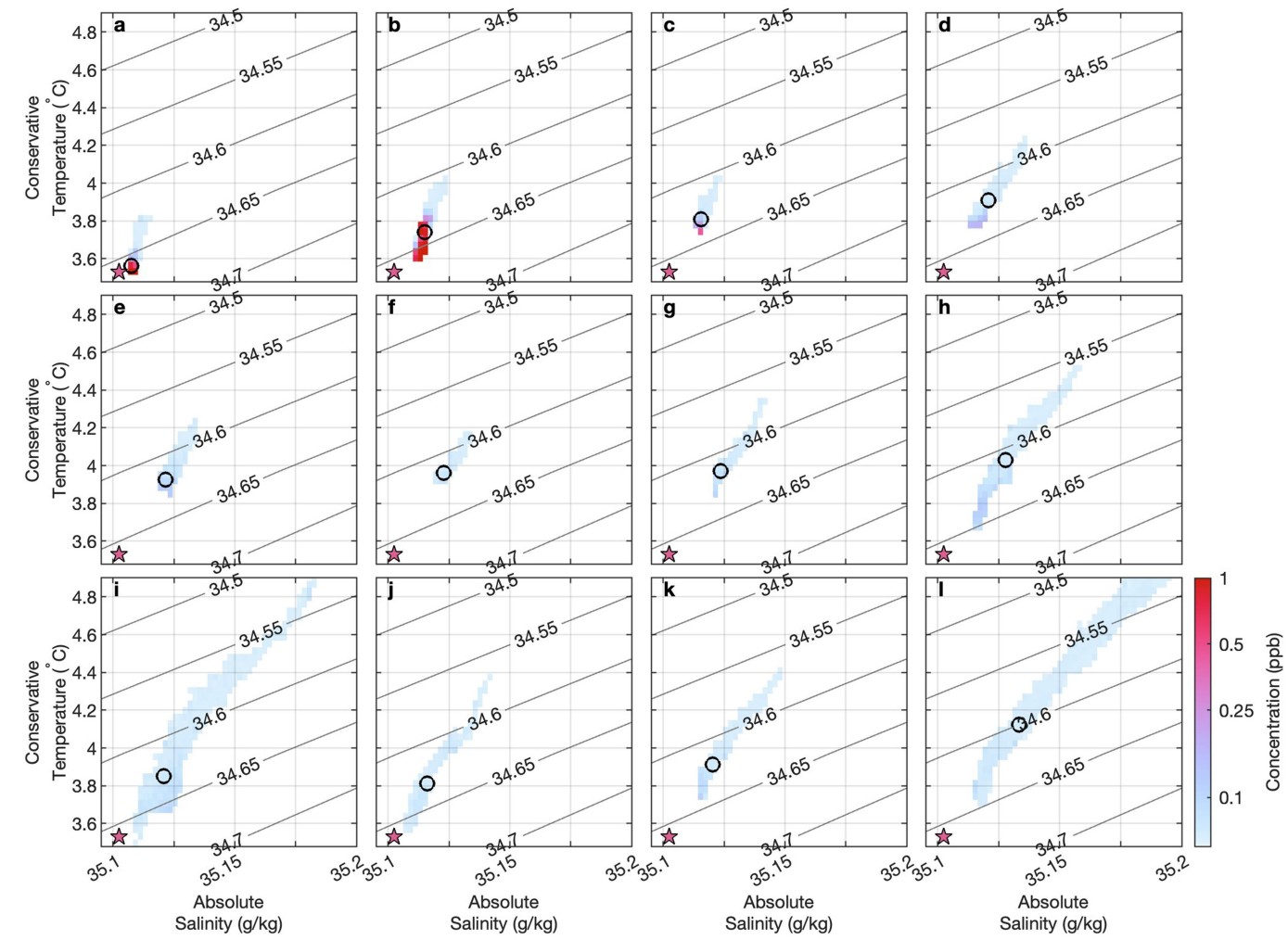

**Extended Data Fig. 6 | Binned dye concentration in conservative temperature-absolute salinity space for each transect.** For each transect ((a)-(l)), all observations are binned in conservative temperature-absolute salinity space, coloured by dye concentration. Contours of potential density anomaly are marked every 0.05 kg m$^{-3}$. The first tracer-weighted density moment is marked with the circle. The potential density anomaly of the dye at the time of release is marked with a pink star.

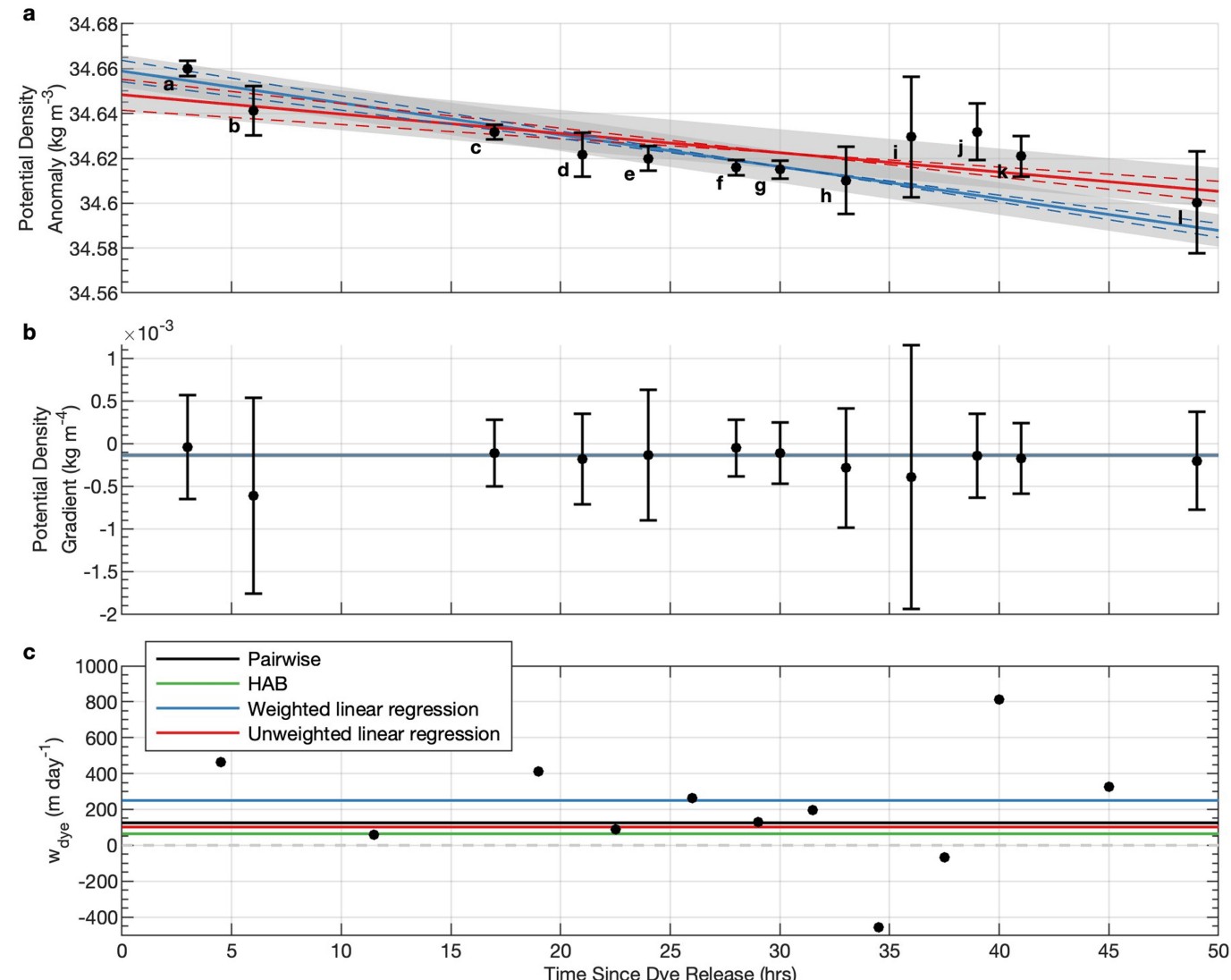

**Extended Data Fig. 7 | Linear regression of the centres of mass of potential density anomaly, the gradient of potential density anomaly over time and estimates of upwelling velocity.** (a) Weighed linear regression (solid blue line) and unweighted linear regression (solid red line) of the centres of mass in potential density anomaly space (black circles). Error bars show standard deviations of dye-weighted density for each transect. The residual standard error of the fit is denoted by the shaded region. Dashed lines show the fits given the standard error on the slope and intercept. The coefficient of determination $R^2$ for the weighted fit is 0.852 and for the unweighted fit is 0.587. (b) Centres of mass in potential density anomaly gradient space (black circles) and standard deviation for each transect (error bars). The solid blue line shows the weighted average of the centres of mass. The standard error of the average is $2.5 \times 10^{-5}$ kg m$^{-4}$ and is smaller than the line width. (c) Estimates of upwelling velocity between subsequent pairs of centres of mass (black circles). Lines denote the upwelling values estimated using different methods: the weighted linear regression (blue), the unweighted linear regression (red), the difference between average height above bottom between the first and last transect (orange), the pairwise estimates in the difference between average height above bottom (green) and the average over all pairwise estimates of the upwelling velocity (black).

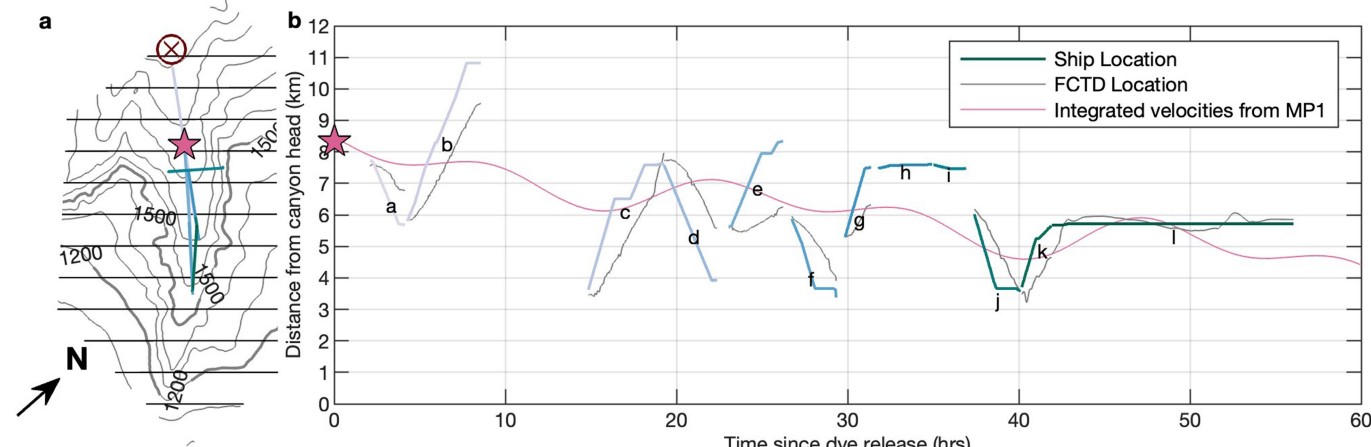

**Extended Data Fig. 8 | Overview of the survey in time and distance along the canyon.** (a) Map of the canyon with bathymetry contoured in grey. Horizontal lines mark the distance along the canyon in kilometres. Coloured lines correspond to the ship's location for each transect. MP1 location is marked by the red cross and the location of the dye release is marked by the pink star. Arrow indicates the direction of north. (b) Distance along the canyon of each transect over time is shown with coloured lines, which correspond to those in panel (a) and letter labels correspond to panels in Extended Data Fig. 5. Thin grey lines show the approximate location of the FCTD at that time based on measured bottom depth. The pink line shows integrated velocity from the MP1 mooring, with a starting position at the dye release location (pink star) as an estimate of the along-canyon position of the dye patch.

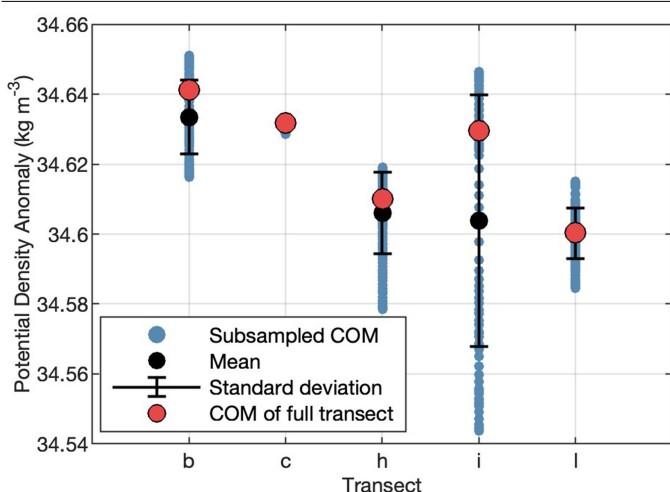

**Extended Data Fig. 9 | Range in the centre of mass for subsections of select transects.** Centre of mass as calculated for subsets of 10 profiles from transects b, c, h, i and l (corresponding to panels in Extended Data Fig. 5). Blue dots indicate the centre of mass (COM) for each subsection, the mean of these is marked by black dots and the standard deviation around this mean is indicated by error bars. COM of the entire transect is shown in red.