## [Peer Review File · Nature]

Manuscript Title: Observations of diapycnal upwelling within a sloping submarine canyon

Editorial Notes:

Redactions – unpublished data

Reviewer Comments & Author Rebuttals

Reviewer Reports on the Initial Version:

Referees' comments:

Referee #1 (Remarks to the Author):

The manuscript presents novel exploration of vigorous mixing in a deep canyon, visualized with fluorescein dye release. To my knowledge, this is the first tracer release study within the bottom boundary layer at such depth (1600-2000m). The study stands out for its ambition and the sheer technological challenge of conducting a dye release study in such an environment. The paper presents compelling observational evidence addressing the issue of “missing” upwelling required to balance the global abyssal water production. The amount of diapycnal dye mixing demonstrated in this experiment is, indeed, impressive for such depths, although not entirely unexpected considering the 0.4 m/s tidal current amplitudes and weak stratification.

Nonetheless, it remains unclear how the results of this study can be generalized beyond the particular submarine canyon in North Atlantic. In fact, even the system under investigation remains substantially unexplored: It is unfortunate that observational challenges did not allow for more comprehensive characterization of the dye patch, particularly in terms of its full 3D extent, second moments, or non-steady-state evolution. Furthermore, it remains unclear to what extent the exceptional upwelling of the dye observed in the bottom boundary layer (BBL) is offset by the downwelling in the stratified mixed layer (SML) above, anticipated to be comparable in magnitude. Without this context, it is difficult to put the observed upwelling rate estimate of 250m/day into perspective and draw meaningful conclusions regarding the role of BBL mixing in driving the global upwelling.

The manuscript is generally well written and logically organized. A few minor issues are addressed in the detailed comments below.

In summary, the paper presents novel and original observations in an extremely challenging environment, offering a rare insight into the abyssal ocean mixing. It holds great promise for spurring further investigations and certainly warrants prompt high-visibility publication. At this stage, however, it offers more questions than answers, making it a potentially better fit for a specialized journal.

Detailed comments:

6: “evidence of this is lacking” – the evidence is not entirely lacking, as demonstrated by the references. “Insufficient”, perhaps?

19: “Tides generate internal wave energy at a rate of ~ 1 TW” – isn’t the wind energy input comparable?

24: The proper reference here is probably “[16]” rather than “[15]”.

30: “To resolve this apparent paradox, upwelling may be confined” – something is missing in this sentence (“it is hypothesized that upwelling...?”)

37: “the definition of a BBL becomes murky” – why is the definition of BBL (or lack thereof) relevant here?

43: “importance of adiabatic and intermittent processes” – this could probably be explained better, particularly clarifying the “intermittent processes”.

45: “all evidence has, to date, been inferred indirectly” – what about Ledwell et al. (2000)?

77: “Previous dye releases have focused on shallow, coastal or lake environments” – this is not quite true, there were several tracer release experiments with the injection depth >500m (including the Ledwell et al., 2000 study cited; also Watson et al., 2013, doi: 10.1038/nature12432, and others).

Granted, none of them were within a deep BBL.

86: “needed to keep the system in steady state” – but this particular system is not in steady state!

113: “in order to maintain this along-canyon flow down the density gradient, there must be mixing across isopycnals” – again, this would only be the case in a steady state, which does not seem to hold here (as shown in Fig 2b).

Fig 3a: A typo in the units of χ .

Fig. 3a-c: These figures would be more effective with the distance as x-axis (perhaps the advective distance in Fig. 3c)

130: “the dye remained ... suggests” – grammar?

131: “The dye was ~50 m” – altitude?

132: “periodic advection of dye into the interior” – why wouldn’t “tidal sloshing” follow the bathymetry?

143: “controlled by the shape of the topography” – this may need to be better explained for the general audience.

421: “ $-1.4 \times 10^{-3} \pm 0.2 \times 10^{-3} \text{ kg m}^{-3} \text{ day}^{-1}$ ” – this value seems to be inconsistent with both the FigA4a and subsequent calculations. Either the numerical value or the units might be wrong.

Referee #2 (Remarks to the Author):

Review of Wynne-Cattanach et al.

Observational evidence of diapycnal upwelling within a sloping submarine canyon

Overview

I enjoyed reading the manuscript from Wynne-Cattanach et al. The paper presents remarkable observations (FCTD and mooring) of diapycnal upwelling in a sloping canyon in the North Atlantic. The authors estimated an upwelling velocity of $250 \pm 75 \text{ m day}^{-1}$ from a dye releasing experiment, which is much higher than previously reported. I believe that this study makes a nice contribution to the ongoing discussions on the role of boundary mixing in upwelling and its impacts on the global overturning circulation. The authors have also clearly discussed the possible sources of uncertainty in their calculations. It is a well-written manuscript which will be of great interest to the readership of Nature. I

recommend publication of this manuscript after a minor revision considering the points raised below.

Minor comments:

1. Increase the size of Figures 1 and 2. With the large number of contours, the figures look very congested.
2. The colormap used to show the dye concentration is not very helpful. Regions of strong concentrations are not obvious especially in Figure 2d, 3a and A5, where isotherms are plotted as thick black lines. Either change the colormap or the color and size of the isotherms.
3. Line 63: "...we use the time-mean buoyancy frequency ...". Please give an indication of the time frame.
4. L67: blue shading in Figure 1a
5. L170: ...significant role in the upwelling...
6. FigA4 caption: "The standard error of the average is smaller than the line width." It would be good to add the value of the standard error in the caption.

Referee #3 (Remarks to the Author):

Observational evidence of diapycnal upwelling within a sloping submarine canyon
Wynne-Cattanach et al.
Nature

Review

This article presents the results of a field campaign conducted in the Rockall Trough. The results of this experiment have been highly anticipated amongst the physical oceanography community as an effort to verify recent theoretical and numerical work highlighting the importance of diapycnal upwelling in thin bottom boundary layers in the abyssal ocean. Significant resources, planning and effort has been put into this field campaign in order to directly measure near-bottom upwelling, and I anticipate that several important results and publications will emerge from it. This article focuses on the tracer release experiment (TRE), with the headline result being an impressive 250m/day average upwelling rate of the tracer, several orders of magnitude larger than the few previous observational TREs performed to date and various other indirect measurements of diapycnal upwelling. The article puts this result into the context of recent literature and discusses its implications.

The article is very well written, accessible to a general audience and the headline result, if robust, is worthy of wide distribution in a journal such as Nature. While I believe the article is ultimately worthy of publication in Nature, I have some concerns about the analysis performed, the robustness of the headline result and the discussion of how that headline result fits into the literature, discussed in more detail below. To me, the results highlight the presence of significant uncertainties in what, the field is

realizing, is a particularly difficult problem (specifically, the relationship between tracer motion and diapycnal transport). While I understand the need to present a relatively clear picture for a general audience, I think the article could do a better job in discussing this complexity and uncertainty. I suggest the article be returned to the authors for major revisions to address these issues.

Major/general comments

1) Complexities in relating the tracer motion to area-integrated "net" diapycnal upwelling: There are complications to interpreting the TRE results in this manner which I believe deserve more discussion, as follows.

- Scales: The present TRE has been conducted at much smaller time and length scales than past TREs (BBTRE, DIMES..). As scales decrease it is not surprising that larger magnitude transport values are seen, as there is less averaging of compensating temporally and spatially varying upwelling and downwelling. One such compensation is that between downwelling in the "stratified mixing layer" (SML) and upwelling within the "bottom boundary layer" (BBL) highlighted in theoretical work. But of course a turbulent flow field can lead to regions of upwelling and downwelling at a range of scales. Thus, any reporting of the contributions of a particular process to the "net" transport must be accompanied by a careful discussion of the scales of averaging. I don't feel that this issue is sufficiently discussed in this article. The authors compare their upwelling rate of 250 m/day to global estimates of AABW upwelling from inverse models (~30Sv, lines 9, 139), transports within MAR canyons averaged over much larger scales (~1000km, multi-year surveys, line 141) and global parameterizations (line 148). The scales of these estimates should be discussed in more detail, as they are not comparable in terms of the amount of averaging performed. In particular, the present estimate may, possibly (see next point), be better thought of as estimate of just BBL upwelling, not the net upwelling within the canyon, and so might not be comparable to ~30Sv of AABW upwelling (the number ~10,000 reported in the abstract could conceivably be increased arbitrarily just by making measurements at smaller scales). The discussion sections of Drake et al. 2022a and Drake et al. 2022b highlight some aspects of this problem and caution against overly simplified interpretations. It boils down to the nature of the time-dependent averaging kernel that the tracer cloud represents.

- "BBL" versus net upwelling: Do the authors take their result as a measure of "BBL" upwelling, or "net" upwelling? It seems clear to me from the tracer transects (Fig. A1), that the tracer remains roughly within the "BBL" (depending on your definition) within the first two transects, after which significant tracer has left the BBL. Fig. A4 suggests that the rate of upwelling decreases with time, as might be expected. Thus, using a linear regression over the whole dataset may yield only a lower bound on the BBL transport, or an over estimate of the net transport. Discussion of this point is important, but is currently absent from the article. Of course, it depends on your definition of the BBL, which also needs to be clarified as the term is used in different contexts throughout the article. Some (in a low-resolution global context) might define the whole canyon to be part of the BBL. On the other hand, microstructure observations are yet to identify a robust near-bottom convergence of the buoyancy flux (e.g. see

companion van Haren manuscript). Reconciliation of these disparate perspectives is a key open question for future research that may be worth mentioning at the end of this article.

- Steady state assumption: How well do the authors expect that their steady state assumption (e.g. in computing the scaling for diapycnal upwelling from the mooring velocities) holds at this location? How representative do the authors think their result is of the true time average? Factors that could influence this include; 1) the spring-neap cycle of turbulence (e.g. see Fig. 4 of St. Laurent et al. 2001) - is this significant in this location and if so, at what phase of this cycle are the measurements taken? 2) large-scale and mesoscale current variability - is information available to know whether the up-canyon velocities observed are typical of the time-mean?

- Hypsometry and whether net upwelling is expected over the entire canyon: I question whether this discussion or material is relevant given the short temporal and small spatial scales involved in the experiment. Hypsometry arguments apply to the degree of compensation between BBL and SML transports associated with variations in slope and circumference, while I think this experiment is more appropriately thought of as a measure of just the BBL component. If the authors do still want to include a discussion of this factor, then it should be made more complete by also including a discussion of other potential drivers of net upwelling. For example, see the decomposition in McDougall and Ferrari 2017 ref 22. Stratification variations could play a significant role.

- Scaling for the diapycnal velocity based on mean up-canyon velocity (e.g. lines 152-154): This approximate number represents an estimate of the net upwelling in the canyon, not BBL upwelling. So is it really comparable to the TRE results?

2) Analysis methods and robustness of result: I found it somewhat difficult to assess the robustness of the main result, given the relatively little detail provided in the methods section. Please provide more details. Specific questions/suggestions:

- As far as I understand it (please clarify), the errors bars in Fig. A4 represent the standard deviation of tracer concentrations across each transect. Is this a relevant measure given it depends on the size of the transect? The sampling uncertainty discussed in the previous section seems more relevant to the uncertainty on the centre of mass than this standard deviation. In particular, this sampling uncertainty should be taken into account in the linear regression and factor into the uncertainty ($\pm 75\text{m/day}$) on the upwelling rate. Is this the case? A similar comment applies to calculation of the mean vertical density gradient.

- As mentioned above, the diapycnal motion of the centre-of-mass is expected to change significantly over the course of the experiment. Perhaps there is not enough data to take this into account in the analysis, but the discussion of the results certainly should.

- Plotting distributions of the tracer in buoyancy space, rather than purely the centre of mass, may assist with answering some of the questions raised above. For example, if the whole tracer distribution moves

as one in the early stages of the experiment, then it is pretty clear that the tracer centre-of-mass is measuring just BBL upwelling.

- Line 428: Given the large uncertainties on the mean vertical density gradient estimate, it would be worth noting how it compares with the average stratification in the ambient waters (within a factor of 2 by my calculations).

- The maximum vertical extent of the transects shown in Figure A1 is roughly from 1800m to 1400m - i.e. about 400m total. The reported upwelling rate of 250m/day, if taken naively in physical space, therefore corresponds to a centre-of-mass movement larger than the vertical extent of the entire domain considered through the 2 days of the experiment. This does not seem plausible. How do the authors explain it? Time variability of the upwelling rate likely plays a role, but is not discussed in the article.

3) "Exchange with the interior" section: this section is brief and its purpose was not clear to me. Is the main point here that the dye has left contact with the bottom at MP1? If so, then (depending on your definition of the BBL) the dye has left the BBL and thus the tracer is no longer measuring just BBL diapycnal transport. I would suggest either removing the section or rewriting to make its purpose clear.

4) Is guidance on the interpretation of the results available from numerical experiments? The discussion of Drake et al. 2022b mentions such experiments and I would be curious to know if they were useful. In particular, could they be used to better constrain sparse-sampling uncertainties for this specific setting and time/space scales?

Minor/specific comments:

- Line 6: The placement of these citations at the end of the sentence is confusing. The citations are support for the first part of the sentence (they are the "observational and theoretical work") not the second.

- Line 13: Thurnherr et al. 2020 (ref 26) would also be appropriate to cite here.

- Line 23: "low observations" -> it's not clear what is meant here. Perhaps "low observed interior mixing"?

- Line 24: This citation to Garrett's comment on the Armi paper doesn't seem appropriate. The Garrett comment is a refutation of the idea stated in this sentence on the basis that the bottom boundary layer's are not well stratified (and has since been proved wrong by observations as well as recent work on 1D BL models such as Callies 2018). A citation to observational papers showing weak mixing in the interior, and/or to the original Armi paper (ref. 30), would seem more appropriate.

- Line 35: There doesn't appear to be any mention of geothermal heating in ref 23 (Mashayek et al.

Nature Communications). Did you instead mean Mashayek et al. (2013, GRL)? Emile-Geay and Madec (2009, e.g. see their Fig. 4) and Adcroft et al. (2001) are also relevant.

- Lines 35-37: There is much recent literature discussing the nature of the BBL in a variety of contexts that would be relevant to cite and perhaps discuss here. For example, the one-dimensional boundary layer theory literature (Garrett papers, Callies 2018) and other threads of literature (Umlauf and Burchard 2011, Holmes and McDougall 2020). In general, I find the introduction is a good summary view of the field for the layman. However, one aspect that I think that is not discussed in sufficient detail is the nature of this BBL (see major comment above).

- Line 41: "chemical" -> "passive"? Since this is in the context of numerical simulations.

- Line 63-64 and Fig. 1 colors: Over what depth range is this calculation performed?

- Line 69: I presume "canyon fork" refers to the mooring in the North West in Fig. 1? Please clarify.

- Fig. 1:

- Please make the panels larger so that the details are easier to see.

- It is very difficult to make out the subcritical and critical regions. Please use a colorbar with a more distinct perceptible scale.

- There is no quantitative scale on the criticality.

- Please add labels to distinguish between moorings MP1 and MP2.

- Line 70: "Daily asymmetry" - I could not easily identify this in the figure. Please clarify.

- Lines 65-74: This discussion of the internal tide and mixing dynamics is not given sufficient context to be understood, particularly for a more general reader. It would be useful to the reader to discuss more of the turbulent dynamics here, possibly on the basis of the accompanying microstructure turbulence observations, before going on to the dye release results.

- Fig. 2:

- This figure is small and of poor quality. Please make the panels larger. I cannot see the details I am interested in, particularly in panel d.

- Please use a colormap for the dye concentration that does not have a dark color as maximum. It is very difficult to distinguish this color from the black of the contour line marking the isotherm of dye release, and hence it is hard to see where there is dye.

- Line 77-78: This statement ignores the BBTRE and DIMES experiments and has no context.

- Line 86: "to keep the system in steady state" - this is confusing, as the system is far from a steady state. A more careful explanation is needed that better reflects the complications that have been the subject of careful recent theoretical and numerical work (e.g. refs. 27, 43 and 44).

- Fig. 3:
 - Please change the colormap as for Fig. 2.
 - Please increase the number of potential density contour lines on Fig. 3d so that one can easily locate the values mentioned in the text (e.g. at lines 95-96).
 - Can the "thick outlines" in Fig. 3d be thickened. It's hard to see which ones you mean.

- Lines 101-104: It's not clear without further context what the inclusion of these two statements is meant to add. Perhaps this could be included along with a more detailed discussion of the microstructure observations earlier on (see comment above on lines 65-74)?

- Line 112-114: This argument assumes a steady state, and there is no discussion of whether this is appropriate. Note that this is a bigger issue for this study than past studies which have averaged over longer time periods. See major comment above.

- Lines 121-122: This is difficult to see in Fig. 2 because of the colormap chosen.

- Lines 123-125: "Dye concentrations were much lower at the mooring than at the FCTD location" - as the values of the concentration at MP1 in Fig. 4 are saturated (i.e. there are many values above the maximum of $1e-3$ ppb), the reader has no way of verifying this statement.

- Line 130: "suggests" -> "suggesting that".

- Line 131: ~50m "thick"?

- Fig. 4: Please change the colormap as for Figs. 2 and 3.

- Lines 140-141: For author information; another point of comparison is the results of Visbeck et al., highlighted in EGU abstract <https://meetingorganizer.copernicus.org/EGU2020/EGU2020-11565.html> but not yet published in the peer-reviewed literature.

- Line 142: Suggest adding "in the mid-Atlantic ridge case" after "slope".

- Line 144-146: Please add a citation to this Monterey canyon work, I presume Kunze et al. 2011 (ref. 7)?

- Lines 145-147: "over the BBL due to hypsometry" - this statement is confusing to me without a specific definition of the BBL. To me, much of the hypsometry argument is about the relative compensation between BBL transport and SML transport for different slopes and circumferences, so shouldn't be applied to the BBL in isolation. Unless your definition of the BBL is very broad (see major comment above).

- Line 164: "these previous studies focused on diabatic and steady flows" - I'm not sure I agree. These

previous studies are based on averages of tidally-modulated flows. They might not resolve the modulations themselves, but nevertheless the mixing fields applied are understood to arise from and include the effects of tidal variability.

- Line 168-169: It's unclear what the authors mean here. Global scalings for what? This point again comes down to averaging. If the measurements of upwelling made here are averaged with large downwelling elsewhere (in time and/or space), then their contribution to the net upwelling is not really possible to compute (it depends on the scale over which the calculation is made). The discussion of Drake et al. 2022a makes a similar point in the context of the MAR observations of ref 26.

- Line 385-386: When the tracer patch is large, isn't sampling less of an issue as the tracer is more homogeneously distributed?

- Line 386-388: It is also possible that MP1, located only in the centre of the canyon, may have missed a large concentrated patch of tracer if it were banked against one wall.

- Line 393-394: There is no discussion of what this number means. How does this uncertainty influence the confidence in your average upwelling rate (see major comment above)?

- Lines 423-425: Can the authors clarify what they mean here. What is the aspect ratio of the overturn in the transect shown in Fig. A1? If it is near 1, then surely horizontal gradients can be just as large as vertical ones, at least on a point-by-point basis.

References:

- Adcroft et al. (2001) Impact of geothermal heating on the global ocean circulation, *Geophysical Research Letters*, 28, 1735-1738, <https://doi.org/10.1029/2000GL012182>

- Emile-Geay & Madec (2009) Geothermal heating, diapycnal mixing and the abyssal circulation, *Ocean Science*, 5, 203-217, <https://doi.org/10.5194/os-5-203-2009>

- Mashayek et al. (2013) The role of the geothermal heat flux in driving the abyssal ocean circulation, *Geophys. Res. Lett.*, 40, 3144-3149, <https://doi.org/10.1002/grl.50640>

- St. Laurent et al. (2001) Buoyancy Forcing by Turbulence above Rough Topography in the Abyssal Brazil Basin, *J. Phys. Oceanogr.*, 31, 3476-3495, [https://doi.org/10.1175/1520-0485\(2001\)031<3476:BFBTAR>2.0.CO;2](https://doi.org/10.1175/1520-0485(2001)031<3476:BFBTAR>2.0.CO;2)

- Callies (2018) Restratification of Abyssal Mixing Layers by Submesoscale Baroclinic Eddies, *J. Phys. Oceanogr.*, 48, 1995-2010, <https://doi.org/10.1175/JPO-D-18-0082.1>

- Umlauf and Burchard (2011) Diapycnal Transport and Mixing Efficiency in Stratified Boundary Layers near Sloping Topography, *J. Phys. Oceanogr.*, 41, 329-345, <https://doi.org/10.1175/2010JPO4438.1>

- Holmes and McDougall (2020) Diapycnal Transport near a Sloping Bottom Boundary, *J. Phys. Oceanogr.*, 50, 3253-3266, <https://doi.org/10.1175/JPO-D-20-0066.1>

- Drake et al. (2022a) Dynamics of Eddyding Abyssal Mixing Layers over Sloping Rough Topography, *Journal of Physical Oceanography*, 52, 3199 - 3219, <https://doi.org/10.1175/JPO-D-22-0009.1>

- Drake et al. (2022b) Diapycnal Displacement, Diffusion, and Distortion of Tracers in the Ocean, *Journal of Physical Oceanography*, 52, 3221 - 3240, <https://doi.org/10.1175/JPO-D-22-0010.1>

Author Rebuttals to Initial Comments:

Response to Referee #1

Referee #1 (Remarks to the Author):

The manuscript presents novel exploration of vigorous mixing in a deep canyon, visualized with fluorescein dye release. To my knowledge, this is the first tracer release study within the bottom boundary layer at such depth (1600-2000m). The study stands out for its ambition and the sheer technological challenge of conducting a dye release study in such an environment. The paper presents compelling observational evidence addressing the issue of “missing” upwelling required to balance the global abyssal water production. The amount of diapycnal dye mixing demonstrated in this experiment is, indeed, impressive for such depths, although not entirely unexpected considering the 0.4 m/s tidal current amplitudes and weak stratification.

Thank you.

Nonetheless, it remains unclear how the results of this study can be generalized beyond the particular submarine canyon in North Atlantic. In fact, even the system under investigation remains substantially unexplored: It is unfortunate that observational challenges did not allow for more comprehensive characterization of the dye patch, particularly in terms of its full 3D extent, second moments, or non-steady-state evolution.

Thank you for this comment. As we have also explained to the Editor, we left some of this important physical characterisation out of our original manuscript because it is contained in a companion manuscript (Naveira Garabato et al 2024) that has not yet been submitted. We are attaching the draft of that manuscript to our revision package for your full context, and also significantly revised our manuscript to better include the requested information. New figures include a time series of the internal tide kinetic energy, a detailed time series demonstrating the dominance of the convective instability mechanism and a demonstration of the steady state of the system on longer-than-tidal timescales.

[REDACTED]

Furthermore, it remains unclear to what extent the exceptional upwelling of the dye observed in the bottom boundary layer (BBL) is offset by the downwelling in the stratified mixed layer (SML) above, anticipated to be comparable in magnitude. Without this context, it is difficult to put the observed upwelling rate estimate of 250m/day into perspective and draw meaningful conclusions regarding the role of BBL mixing in driving the global upwelling.

Thank you very much for this insightful comment. As we now describe in our revised manuscript, i) both dye and tracer measure NET diapycnal transport. And ii) dye and tracer are very different measures of the net diapycnal flow because dye is only detectable for days rather than years as for the tracer. As a result, our near-bottom dye release is strongly dominated by upwelling. We expect the tracer results, which will be submitted in the future, to reflect a closer balance between near-bottom upwelling and interior downwelling.

[REDACTED]

The manuscript is generally well written and logically organized. A few minor issues are addressed in the detailed comments below.

Thank you.

In summary, the paper presents novel and original observations in an extremely challenging environment, offering a rare insight into the abyssal ocean mixing. It holds great promise for spurring further investigations and certainly warrants prompt high-visibility publication. At this stage, however, it offers more questions than answers, making it a potentially better fit for a specialized journal.

Thank you. We agree that questions remain. We believe that our revised manuscript, which contains i) better documentation of the system's near steady state on longer-than-tidal timescales, ii) better error estimates, iii) clearer description that the dye gives net upwelling flow, which is dominated by upwelling because of its proximity to the bottom; iv) better description of the full physical system in which the upwelling is occurring and v) more convincing generalisability arguments, will convince reviewers and the Editor that our results do in fact belong in Nature. We believe that our first-ever direct measurement of upwelling along a sloping bottom and its contextualisation with supporting measurements represent the beginning of a paradigm shift in our understanding of the role of mixing in the ocean that should be reported to the widest possible audience.

Detailed comments:

6: "evidence of this is lacking" – the evidence is not entirely lacking, as demonstrated by the references. "Insufficient", perhaps?

We agree this was unclear. The sentence has been rephrased as suggested.

19: “Tides generate internal wave energy at a rate of ~1 TW” – isn’t the wind energy input comparable?

Thank you. We have revised the manuscript to state “The wind and the tides”

24: The proper reference here is probably “[16]” rather than “[15]”.

Thank you for spotting this mistake. We have cited Wunsch and Ferrari 2004 (ref. 1) instead as this discusses microstructure observations in the context of global circulation.

30: “To resolve this apparent paradox, upwelling may be confined” – something is missing in this sentence (“it is hypothesized that upwelling...”?)

We agree. The sentence has been edited to include this suggestion.

37: “the definition of a BBL becomes murky” – why is the definition of BBL (or lack thereof) relevant here?

The term “bottom boundary layer” has been used frequently in the literature, but with different definitions. A result of this experiment (discussed in more detail in the companion work by Naveira Garabato et al.) is that the near-boundary region involves complex processes that have not been accounted for in previous theoretical work. We have expanded this section to discuss the previous definitions of the bottom boundary layer to contextualise this result.

43: “importance of adiabatic and intermittent processes” – this could probably be explained better, particularly clarifying the “intermittent processes”.

We have removed this sentence and instead refer more explicitly to the time-dependent and 3D processes occurring near sloping topography.

45: “all evidence has, to date, been inferred indirectly” – what about Ledwell et al. (2000)?

Our interpretation of the Ledwell et al result is that while tracer appeared to move upwards along the boundary, the one dimensional model used did not allow them to determine whether or not this was in fact diapycnal upwelling and therefore believe that referring to this as indirect evidence is justified. We have included more discussion of previous tracer experiments in the revised manuscript.

77: “Previous dye releases have focused on shallow, coastal or lake environments” – this is not quite true, there were several tracer release experiments with the injection depth >500m (including the Ledwell et al., 2000 study cited; also Watson et al., 2013, doi: 10.1038/nature12432, and others). Granted, none of them were within a deep BBL.

True, thanks. While there have been previous tracer releases in the deep ocean, the distinction between dye and chemical tracer deserves clarification. We have rewritten this section to include the suggested examples, and now highlight the differences in sampling and scales of tracer and dye.

86: “needed to keep the system in steady state” – but this particular system is not in steady state!

Thank you for this comment, which highlights our failure to make a central point in the original manuscript: that the system is in fact in near-steady state on timescales longer than a tidal period, but that it is the intrinsic time dependence of the internal tides themselves that provides the key to the upwelling mechanism. We have revised and expanded this discussion to make this point more clearly, and have added two additional figures to demonstrate i) the crucial role of the tides in causing the convective instability but that ii) the tidally-averaged system is in fact in near steady state.

113: “in order to maintain this along-canyon flow down the density gradient, there must be mixing across isopycnals” – again, this would only be the case in a steady state, which does not seem to hold here (as shown in Fig 2b).

Thank you for pointing out that the discussion of this calculation was unclear. As mentioned above, we have added a figure to the extended data items (Fig ED3) which demonstrates that on longer than tidal timescales, the temperature field is approximately steady. Therefore, the time-averaged up-canyon flow must be associated with diapycnal upwelling. We have expanded on this point in the text and hope that we have been able to clarify why this assumption of steady-state on these timescales is valid, despite the clear importance of tidally driven mixing.

Fig 3a: A typo in the units of χ .

Thank you for spotting this. The units have been corrected to $^{\circ}\text{C}^2/\text{s}$ in the figure.

Fig. 3a-c: These figures would be more effective with the distance as x-axis (perhaps the advective distance in Fig. 3c)

Due to Doppler shifting the observations of the dye patch become distorted when plotted with distance on the x-axis. Therefore, we have chosen to keep the figures with the x-axis as time, but have included a scale bar on the first two panels to provide an estimate of the horizontal scale of the patch.

130: “the dye remained ... suggests” – grammar?

“Suggests” has been changed to “suggesting” to fix the grammar.

131: “The dye was ~50 m” – altitude?

Added “above the bottom”.

132: “periodic advection of dye into the interior” – why wouldn’t “tidal sloshing” follow the bathymetry?

Although the velocities approximately follow the bathymetry (they are slightly steeper during upcanyon flow and shallower during downcanyon flow), the dye is transported across isopycnals during upcanyon flow but along isopycnals and horizontally away from the boundary, in part due to convergence in the velocity parallel to the slope. We have significantly revised this section to make this clear in the revised manuscript.

143: “controlled by the shape of the topography” – this may need to be better explained for the general audience.

We have added a couple of sentences to clarify what this means.

421: “ $-1.4 \times 10^{-3} \pm 0.2 \times 10^{-3} \text{ kg m}^{-3} \text{ day}^{-1}$ ” – this value seems to be inconsistent with both the FigA4a and subsequent calculations. Either the numerical value or the units might be wrong.

Thank you for noticing this, the units were incorrect and for the value previously given should have been $\text{kg m}^{-3} \text{ hr}^{-1}$. The value and the units have now been changed to be in $\text{kg m}^{-3} \text{ day}^{-1}$.

Response to Referee #2

Referee #2 (Remarks to the Author):

Review of Wynne-Cattanach et al.

Observational evidence of diapycnal upwelling within a sloping submarine canyon

Overview

I enjoyed reading the manuscript from Wynne-Cattanach et al. The paper presents remarkable observations (FCTD and mooring) of diapycnal upwelling in a sloping canyon in the North Atlantic. The authors estimated an upwelling velocity of 250 ± 75 m day⁻¹ from a dye releasing experiment, which is much higher than previously reported. I believe that this study makes a nice contribution to the on-going discussions on the role of boundary mixing in upwelling and its impacts on the global overturning circulation. The authors have also clearly discussed the possible sources of uncertainty in their calculations. It is a well-written manuscript which will be of great interest to the readership of Nature. I recommend publication of this manuscript after a minor revision considering the points raised below.

Thank you.

Minor comments:

1. Increase the size of Figures 1 and 2. With the large number of contours, the figures look very congested.

The size of both figures 1 and 2 has been increased.

2. The colormap used to show the dye concentration is not very helpful. Regions of strong concentrations are not obvious especially in Figure 2d, 3a and A5, where isotherms are plotted as thick black lines. Either change the colormap or the color and size of the isotherms.

The colourmap has been changed to one that does not end in such a dark colour. We believe this is a significant improvement over the previous colourmap and so thank you very much for this comment.

3. Line 63: "...we use the time-mean buoyancy frequency ...". Please give an indication of the time frame.

We have added that the CTD casts we collected over the course of the 5 weeks of the experiment.

4. L67: blue shading in Figure 1a

This has been added.

5. L170: ...significant role in the upwelling...

"Significantly" has been changed to significant.

6. FigA4 caption: "The standard error of the average is smaller than the line width." It would be good to add the value of the standard error in the caption.

The error ($2.5 \times 10^{-5} \text{ kg m}^{-4}$) has been added to the caption.

Response to Referee #3

Referee #3 (Remarks to the Author):

Observational evidence of diapycnal upwelling within a sloping submarine canyon
Wynne-Cattanach et al.
Nature

Review

This article presents the results of a field campaign conducted in the Rockall Trough. The results of this experiment have been highly anticipated amongst the physical oceanography community as an effort to verify recent theoretical and numerical work highlighting the importance of diapycnal upwelling in thin bottom boundary layers in the abyssal ocean. Significant resources, planning and effort has been put into this field campaign in order to directly measure near-bottom upwelling, and I anticipate that several important results and publications will emerge from it. This article focuses on the tracer release experiment (TRE), with the headline result being an impressive 250m/day average upwelling rate of the tracer, several orders of magnitude larger than the few previous observational TREs performed to date and various other indirect measurements of diapycnal upwelling. The article puts this result into the context of recent literature and discusses its implications.

The article is very well written, accessible to a general audience and the headline result, if robust, is worthy of wide distribution in a journal such as Nature. While I believe the article is ultimately worthy of publication in Nature, I have some concerns about the analysis performed, the robustness of the headline result and the discussion of how that headline result fits into the literature, discussed in more detail below. To me, the results highlight the presence of significant uncertainties in what, the field is realizing, is a particularly difficult problem (specifically, the relationship between tracer motion and diapycnal transport). While I understand the need to present a relatively clear picture for a general audience, I think the article could do a better job in discussing this complexity and uncertainty. I suggest the article be returned to the authors for major revisions to address these issues.

Thank you. We have addressed the major and minor issues below.

Major/general comments

1) Complexities in relating the tracer motion to area-integrated "net" diapycnal upwelling: There are complications to interpreting the TRE results in this manner which I believe deserve more discussion, as follows.

- Scales: The present TRE has been conducted at much smaller time and length scales than past TREs (BBTRE, DIMES..). As scales decrease it is not surprising that larger magnitude transport values are seen, as there is less averaging of compensating temporally and spatially varying upwelling and downwelling. One such compensation is that between downwelling in the "stratified mixing layer" (SML) and upwelling within the "bottom boundary layer" (BBL) highlighted in theoretical work. But of course a turbulent flow field can lead to regions of upwelling and downwelling at a range of scales. Thus, any reporting of the contributions of a particular process to the "net" transport must be accompanied by a careful discussion of the scales of averaging. I don't feel that this issue

is sufficiently discussed in this article. The authors compare their upwelling rate of 250 m/day to global estimates of AABW upwelling from inverse models (~30Sv, lines 9, 139), transports within MAR canyons averaged over much larger scales (~1000km, multi-year surveys, line 141) and global parameterizations (line 148). The scales of these estimates should be discussed in more detail, as they are not comparable in terms of the amount of averaging performed. In particular, the present estimate may, possibly (see next point), be better thought of as estimate of just BBL upwelling, not the net upwelling within the canyon, and so might not be comparable to ~30Sv of AABW upwelling (the number ~10,000 reported in the abstract could conceivably be increased arbitrarily just by making measurements at smaller scales). The discussion sections of Drake et al. 2022a and Drake et al. 2022b highlight some aspects of this problem and caution against overly simplified interpretations. It boils down to the nature of the time-dependent averaging kernel that the tracer cloud represents.

Thank you; this is a key point that was also brought up by another reviewer. We agree that we failed to adequately describe i) the fact that all tracers reflect the NET integrated upwelling, and ii) the difference in scales of this (dye) experiment compared to previous (SF6-type tracer) experiments was not sufficiently discussed. Because dye is only detectable for days rather than months/years, a near-bottom dye release preferentially samples the upwelling regime in the bottom ~100 m. We have included discussion of the difference between a dye release such as this and a long term tracer release, particularly the time and spatial scales. As suggested, the summary section has been expanded to highlight how previous estimates of upwelling are not comparable due to the different scales.

[REDACTED]

- "BBL" versus net upwelling: Do the authors take their result as a measure of "BBL" upwelling, or "net" upwelling? It seems clear to me from the tracer transects (Fig. A1), that the tracer remains roughly within the "BBL" (depending on your definition) within the first two transects, after which significant tracer has left the BBL. Fig. A4 suggests that the rate of upwelling decreases with time, as might be expected. Thus, using a linear regression over the whole dataset may yield only a lower bound on the BBL transport, or an over estimate of the net transport. Discussion of this point is important, but is currently absent from the article. Of course, it depends on your definition of the BBL, which also needs to be clarified as the term is used in different contexts throughout the article. Some (in a low-resolution global context) might define the whole canyon to be part of the BBL. On the other hand, microstructure observations are yet to identify a robust near-bottom convergence of the buoyancy flux (e.g. see companion van Haren manuscript). Reconciliation of these disparate perspectives is a key open question for future research that may be worth mentioning at the end of this article.

Thank you: you are making three important points here. To your first point regarding net versus total upwelling, please see our above response to your and the other reviewer's comments. We now make it clear that i) dye and tracer both measure net upwelling; that ii) the proximity to the bottom of the dye cloud while it was detectable cause it to be dominated by upwelling and that iii) results from the companion manuscript Naveira Garabato et al (2024), showing that independent estimates of diapycnal transport demonstrate upwelling in the bottom ~100 m with downwelling aloft. We make each of these points much more clearly now in our revised manuscript.

To your second point, we have completely revised our treatment of the BBL, completely avoiding reference to a BBL.

To your third point, our moored chi estimates do in fact support upwelling, as documented in the companion paper Naveira Garabato et al 2024; we now mention this more clearly in our own manuscript. However, we completely agree with you that there are outstanding questions. We mention these in the summary as you have requested.

- Steady state assumption: How well do the authors expect that their steady state assumption (e.g. in computing the scaling for diapycnal upwelling from the mooring velocities) holds at this location? How representative do the authors think their result is of the true time average? Factors that could influence this include; 1) the spring-neap cycle of turbulence (e.g. see Fig. 4 of St. Laurent et al. 2001) - is this significant in this location and if so, at what phase of this cycle are the measurements taken? 2) large-scale and mesoscale current variability - is information available to know whether the up-canyon velocities observed are typical of the time-mean?

Thank you. We agree that insufficient evidence for the system's steady state was provided in the original manuscript. The key point is that while the time variability of the internal tides is crucial to the upwelling mechanism, the system is in fact in quasi steady state on time scales longer than a tidal period. New figures include evidence that the steady state assumption is valid, documentation of the magnitude of the spring/neap cycle and long-term estimates of the upwelling from the moorings. Regarding mesoscale variability, we chose the experiment's location in part because of minimal mesoscale flows, which we in fact did observe and now document and discuss. We have expanded our discussion to address each of these points carefully.

- Hypsometry and whether net upwelling is expected over the entire canyon: I question whether this discussion or material is relevant given the short temporal and small spatial scales involved in the experiment. Hypsometry arguments apply to the degree of compensation between BBL and SML transports associated with variations in slope and circumference, while I think this experiment is more appropriately thought of as a measure of just the BBL component. If the authors do still want to include a discussion of this factor, then it should be made more complete by also including a discussion of other potential drivers of net upwelling. For example, see the decomposition in McDougall and Ferrari 2017 ref 22. Stratification variations could play a significant role.

Thank you. We agree and have removed this section which was not central to our results.

- Scaling for the diapycnal velocity based on mean up-canyon velocity (e.g. lines 152-154): This approximate number represents an estimate of the net upwelling in the canyon, not BBL upwelling. So is it really comparable to the TRE results?

Please see our response above. We now include an explicit discussion that the dye measures net upwelling, but that its shorter time and space scales while it is detectable (compared to a long-term tracer) cause it to remain in the region dominated by upwelling during our measurements.

[REDACTED]

Therefore, the dye-inferred upwelling is in fact comparable to the moored estimate, in the same sense as previous results comparing inflow and diapycnal outflow in the Brazil Basin (e.g. Hogg, St. Laurent et al 2001).

2) Analysis methods and robustness of result: I found it somewhat difficult to assess the robustness of the main result, given the relatively little detail provided in the methods section. Please provide more details. Specific questions/suggestions:

Thank you. As described in detail below, we have significantly expanded our description of the techniques and their errors in the methods section.

- As far as I understand it (please clarify), the errors bars in Fig. A4 represent the standard deviation of tracer concentrations across each transect. Is this a relevant measure given it depends on the size of the transect? The sampling uncertainty discussed in the previous section seems more relevant to the uncertainty on the centre of mass than this standard deviation. In particular, this sampling uncertainty should be taken into account in the linear regression and factor into the uncertainty (+75m/day) on the upwelling rate. Is this the case? A similar comment applies to calculation of the mean vertical density gradient.

You are correct that the error bars represent the standard deviations across each transect. For the transects where it is possible to perform the analysis and compare, the standard deviation of the COM estimated through subsampling (Fig ED6) approximately scale like the standard deviation of the whole transect (Fig ED7, panel (a)). Since the analysis done for Fig ED6 cannot be done for all transects, we use the standard deviation for the whole transect as an estimate of the sampling error. We are aware that there are cases where this method may not accurately represent the true sampling error, particularly for short transects where the standard deviation is small but the centre of mass is likely to be more different than observed. However, we have no alternative way to estimate the error. We have added discussion of these issues to the methods section.

- As mentioned above, the diapycnal motion of the centre-of-mass is expected to change significantly over the course of the experiment. Perhaps there is not enough data to take this into account in the analysis, but the discussion of the results certainly should.

Thank you. We agree that this was not discussed in sufficient detail in the original manuscript. The revised methods section includes discussion of estimating the diapycnal velocity as the average over all pairwise estimates of the velocity and Figure ED7 now shows that there is some time variability. We added discussion of this to the text with the caveat that our error bars are too large to determine the true variability.

- Plotting distributions of the tracer in buoyancy space, rather than purely the centre of mass, may assist with answering some of the questions raised above. For example, if the whole tracer distribution moves as one in the early stages of the experiment, then it is pretty clear that the tracer centre-of-mass is measuring just BBL upwelling.

Figure ED8 plots the whole tracer distribution for each transect in density space. We do see that the whole tracer patch moved upwards at the beginning of the experiment. Determining whether the later transects aren't measuring just near-bottom upwelling is difficult given the sampling error, but may indeed include some downwelling during the later parts of the experiment. We have included this in the text.

- Line 428: Given the large uncertainties on the mean vertical density gradient estimate, it would be worth noting how it compares with the average stratification in the ambient waters (within a factor of 2 by my calculations).

The density gradient without weighting by dye is 2.0×10^{-4} for the full record. This has been added to the methods.

- The maximum vertical extent of the transects shown in Figure A1 is roughly from 1800m to 1400m - i.e. about 400m total. The reported upwelling rate of 250m/day, if taken naively in physical space, therefore corresponds to a centre-of-mass movement larger than the vertical extent of the entire domain considered through the 2 days of the experiment. This does not seem plausible. How do the authors explain it? Time variability of the upwelling rate likely plays a role, but is not discussed in the article.

We agree that this value seems to disagree with the vertical extent of the dye patch in physical space. Possible explanations include the treatment of the time variability as you mention, the weighting due to the error and the choice of the average density gradient to convert from the rate of change of density to a vertical velocity. In the revised methods section we now describe four different estimates of the upwelling rate that use various levels of error and averaging which give a range that aligns more with the vertical extent observed. Discussion of the caveats of each method are also included.

3) "Exchange with the interior" section: this section is brief and its purpose was not clear to me. Is the main point here that the dye has left contact with the bottom at MP1? If so, then (depending on your definition of the BBL) the dye has left the BBL and thus the tracer is no longer measuring just BBL diapycnal transport. I would suggest either removing the section or rewriting to make its purpose clear.

Thank you. Your comment indicates that we failed to effectively make a central point of our manuscript: namely, that our results are inconsistent with 1D models. The injection of the dye into the interior is the primary evidence for this conclusion. We've expanded and rewritten this section to make its significance more clear.

4) Is guidance on the interpretation of the results available from numerical experiments? The discussion of Drake et al. 2022b mentions such experiments and I would be curious to know if they were useful. In particular, could they be used to better constrain sparse-sampling uncertainties for this specific setting and time/space scales?

We had the same instinct, and were in fact planning on including numerical results in this paper both for interpretation as well as assessment of the sampling errors. However, for a variety of reasons a numerical model that adequately characterises all of the physical processes in our system has not emerged - in spite of two different groups on our team. Important details such as the phasing of the turbulence with respect to the internal tide displacements have proven elusive. A detailed numerical study and comparison to the dye experiment will therefore have to be the subject of a future study.

Minor/specific comments:

- Line 6: The placement of these citations at the end of the sentence is confusing. The citations are support for the first part of the sentence (they are the "observational and theoretical work") not the second.

We agree; the citations have been moved to after "observational and theoretical work".

- Line 13: Thurnherr et al. 2020 (ref 26) would also be appropriate to cite here.

We agree; the reference has been added.

- Line 23: "low observations" -> it's not clear what is meant here. Perhaps "low observed interior mixing"?

The phrasing here was confusing; the suggested clarification has been added.

- Line 24: This citation to Garrett's comment on the Armi paper doesn't seem appropriate. The Garrett comment is a refutation of the idea stated in this sentence on the basis that the bottom boundary layer's are not well stratified (and has since been proved wrong by observations as well as recent work on 1D BL models such as Callies 2018). A citation to observational papers showing weak mixing in the interior, and/or to the original Armi paper (ref. 30), would seem more appropriate.

Thank you for spotting this mistake. We have cited Wunsch and Ferrari 2004 (ref. 1) instead as this discusses microstructure observations in the context of global circulation.

- Line 35: There doesn't appear to be any mention of geothermal heating in ref 23 (Mashayek et al. Nature Communications). Did you instead mean Mashayek et al. (2013, GRL)? Emile-Geay and Madec (2009, e.g. see their Fig. 4) and Adcroft et al. (2001) are also relevant.

This was the incorrect reference; thank you for noticing. We have added the correct Mashayek paper. Due to the limited number of references available in the main body of the paper we were unable to include the earlier suggested works.

- Lines 35-37: There is much recent literature discussing the nature of the BBL in a variety of contexts that would be relevant to cite and perhaps discuss here. For example, the one-dimensional boundary layer theory literature (Garrett papers, Callies 2018) and other threads of literature (Umlauf and Burchard 2011, Holmes and McDougall 2020). In general, I find the introduction is a good summary view of the field for the layman. However, one aspect that I think that is not discussed in sufficient detail is the nature of this BBL (see major comment above).

We have added more detail to the introduction to discuss the definitions of the BBL in various contexts, as suggested.

- Line 41: "chemical" -> "passive"? Since this is in the context of numerical simulations.

We have changed the phrasing to "passive".

- Line 63-64 and Fig. 1 colors: Over what depth range is this calculation performed?

The time average stratification profile estimated from CTD profiles collected over the course of the experiment was interpolated onto the bathymetry to obtain an estimate of the stratification at each location. The time period over which the stratification was calculated has been added to the caption and text; we hope this edit makes this more clear.

- Line 69: I presume "canyon fork" refers to the mooring in the North West in Fig. 1? Please clarify.

This sentence has been removed in the revised manuscript; however, labels have been added to the figure as suggested below.

- Fig. 1:
- Please make the panels larger so that the details are easier to see.
- It is very difficult to make out the subcritical and critical regions. Please use a colorbar with a more distinct perceptible scale.
- There is no quantitative scale on the criticality.
- Please add labels to distinguish between moorings MP1 and MP2.

The figure has been made larger so that details are now clearer. We have enhanced the colours in the colourmap to highlight the subcritical and critical regions. The colour bar has both a qualitative (sub/super critical) and quantitative axis (log of the criticality with values 0.1, 1 and 10 marked). The labels for the moorings have been added.

- Line 70: "Daily asymmetry" - I could not easily identify this in the figure. Please clarify.

This sentence has been removed from the revised manuscript.

- Lines 65-74: This discussion of the internal tide and mixing dynamics is not given sufficient context to be understood, particularly for a more general reader. It would be useful to the reader to discuss more of the turbulent dynamics here, possibly on the basis of the accompanying microstructure turbulence observations, before going on to the dye release results.

We have included new extended data items which show other measurements from other moorings deployed during the experiment and have expanded the discussion of the internal tide in the text with references to these observations.

- Fig. 2:
- This figure is small and of poor quality. Please make the panels larger. I cannot see the details I am interested in, particularly in panel d.
- Please use a colormap for the dye concentration that does not have a dark color as maximum. It is very difficult to distinguish this color from the black of the contour line marking the isotherm of dye release, and hence it is hard to see where there is dye.

The figure has been enlarged and the colourmap of the dye has been changed such that the large values are not dark and do not interfere with the isotherms.

- Line 77-78: This statement ignores the BBTRE and DIMES experiments and has no context.

This section has been expanded to clarify that dye releases are distinct from tracer releases, in particular that the frequency of the observations allows for quantification of the watermass transformation over smaller spatial and temporal scales. Previous dye releases have not been completed in an environment such as this and we hope that with the additional discussion, this point is no longer out of context.

- Line 86: "to keep the system in steady state" - this is confusing, as the system is far from a steady state. A more careful explanation is needed that better reflects the complications

that have been the subject of careful recent theoretical and numerical work (e.g. refs. 27, 43 and 44).

This phrase has been removed and the discussion of the systems steady-state on greater than tidal timescales has been expanded (See response to major comment about the steady state assumption).

- Fig. 3:

- Please change the colormap as for Fig. 2.

Done.

- Please increase the number of potential density contour lines on Fig. 3d so that one can easily locate the values mentioned in the text (e.g. at lines 95-96).

Done.

- Can the "thick outlines" in Fig. 3d be thickened. It's hard to see which ones you mean.

Done.

- Lines 101-104: It's not clear without further context what the inclusion of these two statements is meant to add. Perhaps this could be included along with a more detailed discussion of the microstructure observations earlier on (see comment above on lines 65-74)?

We agree that these sentences are unnecessary here and have moved this discussion to earlier in the manuscript where we describe the details of the dynamics in this canyon.

- Line 112-114: This argument assumes a steady state, and there is no discussion of whether this is appropriate. Note that this is a bigger issue for this study than past studies which have averaged over longer time periods. See major comment above.

See response to major comment about the steady state assumption.

- Lines 121-122: This is difficult to see in Fig. 2 because of the colormap chosen.

The colourmap and colourmap limits have been changed so that the second observation of the dye is more obvious.

- Lines 123-125: "Dye concentrations were much lower at the mooring than at the FCTD location" - as the values of the concentration at MP1 in Fig. 4 are saturated (i.e. there are many values above the maximum of $1e-3$ ppb), the reader has no way of verifying this statement.

The colourmap limits for the dye concentration at MP1 have been changed to more clearly demonstrate the difference in the concentrations between the two locations.

- Line 130: "suggests" -> "suggesting that".

This has been fixed.

- Line 131: ~50m "thick"?

Added "above the bottom".

- Fig. 4: Please change the colormap as for Figs. 2 and 3.

The colourmap has been changed.

- Lines 140-141: For author information; another point of comparison is the results of Visbeck et al., highlighted in EGU abstract <https://meetingorganizer.copernicus.org/EGU2020/EGU2020-11565.html> but not yet published in the peer-reviewed literature.

Thank you for bringing this work to our attention.

- Line 142: Suggest adding "in the mid-Atlantic ridge case" after "slope".

Done.

- Line 144-146: Please add a citation to this Monterey canyon work, I presume Kunze et al. 2011 (ref. 7)?

Thank you for pointing out that the citation was missing; it has been added.

- Lines 145-147: "over the BBL due to hypsometry" - this statement is confusing to me without a specific definition of the BBL. To me, much of the hypsometry argument is about the relative compensation between BBL transport and SML transport for different slopes and circumferences, so shouldn't be applied to the BBL in isolation. Unless your definition of the BBL is very broad (see major comment above).

See above responses to major comments. We have edited this section to remove the reference to the BBL and emphasise that this estimate is a net upwelling over a larger volume than our study.

- Line 164: "these previous studies focused on diabatic and steady flows" - I'm not sure I agree. These previous studies are based on averages of tidally-modulated flows. They might not resolve the modulations themselves, but nevertheless the mixing fields applied are understood to arise from and include the effects of tidal variability.

We have rephrased this section to say that the models used parametrised tidal mixing while we are able to observe the water mass transformation directly.

- Line 168-169: It's unclear what the authors mean here. Global scalings for what? This point again comes down to averaging. If the measurements of upwelling made here are averaged with large downwelling elsewhere (in time and/or space), then their contribution to the net upwelling is not really possible to compute (it depends on the scale over which the calculation is made). The discussion of Drake et al. 2022a makes a similar point in the context of the MAR observations of ref 26.

This section now states that our work demonstrates upwelling near the bottom that is larger than previous estimates due to the more focused experiment design and provides direct evidence of near bottom upwelling that theoretical works predicted.

- Line 385-386: When the tracer patch is large, isn't sampling less of an issue as the tracer is more homogeneously distributed?

Although the dye may have been more homogeneously distributed, the sampling for some of the later transects was limited in temperature space, therefore biasing the center of mass. A paragraph has been added to the manuscript that describes this in detail.

- Line 386-388: It is also possible that MP1, located only in the centre of the canyon, may have missed a large concentrated patch of tracer if it were banked against one wall.

This is true; this clarification has been added.

- Line 393-394: There is no discussion of what this number means. How does this uncertainty influence the confidence in your average upwelling rate (see major comment above)?

This number has been removed and replaced with a more detailed discussion of why the standard deviation was chosen as the uncertainty in the linear regression. (See response to major comment).

- Lines 423-425: Can the authors clarify what they mean here. What is the aspect ratio of the overturn in the transect shown in Fig. A1? If it is near 1, then surely horizontal gradients can be just as large as vertical ones, at least on a point-by-point basis.

Thank you, you are correct; we have revised our statement accordingly. We crudely estimate from Figure ED1 that the impact on our vertical velocity estimates of this approximation is O(5%): 1) we estimate an extreme isopycnal slope from Figure ED1 as $dz/dx=(dz/dt)/u = 0.1$, where the maximum value of dz/dt is 200 m / (2 hours) and $u = 0.25$ m/s. 2) If this slope were always present, our estimates would indeed be off by 10% owing to the neglect of horizontal gradients. However, the slopes are only this large for a small fraction of the tidal cycle. Conservatively assuming that the slope goes from 0 to 0.1 sinusoidally, we estimate the error from assuming purely horizontal gradients over a tidal cycle as we do would be 5%, well smaller than our other sources of error. In reality, because the borelike features are non-sinusoidal, the true errors are smaller.

References:

- Adcroft et al. (2001) Impact of geothermal heating on the global ocean circulation, Geophysical Research Letters, 28, 1735-1738, <https://doi.org/10.1029/2000GL012182>

- Emile-Geay & Madec (2009) Geothermal heating, diapycnal mixing and the abyssal circulation, Ocean Science, 5, 203-217, <https://doi.org/10.5194/os-5-203-2009>

- Mashayek et al. (2013) The role of the geothermal heat flux in driving the abyssal ocean circulation, Geophys. Res. Lett., 40, 3144-3149, <https://doi.org/10.1002/grl.50640>

- St. Laurent et al. (2001) Buoyancy Forcing by Turbulence above Rough Topography in the Abyssal Brazil Basin, J. Phys. Oceanogr., 31, 3476-3495, [https://doi.org/10.1175/1520-0485\(2001\)031<3476:BFBTAR>2.0.CO;2](https://doi.org/10.1175/1520-0485(2001)031<3476:BFBTAR>2.0.CO;2)

- Callies (2018) Restratification of Abyssal Mixing Layers by Submesoscale Baroclinic Eddies, J. Phys. Oceanogr., 48, 1995-2010, <https://doi.org/10.1175/JPO-D-18-0082.1>

- Umlauf and Burchard (2011) Diapycnal Transport and Mixing Efficiency in Stratified Boundary Layers near Sloping Topography, J. Phys. Oceanogr., 41, 329-345, <https://doi.org/10.1175/2010JPO4438.1>

- Holmes and McDougall (2020) Diapycnal Transport near a Sloping Bottom Boundary, J. Phys. Oceanogr., 50, 3253-3266, <https://doi.org/10.1175/JPO-D-20-0066.1>
- Drake et al. (2022a) Dynamics of Eddying Abyssal Mixing Layers over Sloping Rough Topography, Journal of Physical Oceanography, 52, 3199 - 3219, <https://doi.org/10.1175/JPO-D-22-0009.1>
- Drake et al. (2022b) Diapycnal Displacement, Diffusion, and Distortion of Tracers in the Ocean, Journal of Physical Oceanography, 52, 3221 - 3240, <https://doi.org/10.1175/JPO-D-22-0010.1>

Reviewer Reports on the First Revision:

Referees' comments:

Referee #1 (Remarks to the Author):

In the revised manuscript, the authors addressed most of my comments and suggestions.

However, the issue of the steady-state assumption, which is critical for potential generalization of the limited observations, remains.

The authors claim that the system is “in near-steady state on timescales longer than a tidal period” and that the temperature field is “relatively time-invariant”. Figures ED2 and ED3 are presented as evidence of this claim. To me, the figures demonstrate the opposite: Upwelling velocity (Fig. ED2b) shows ample variations of 100-300 m/day on time scales of 1-10 days. Similarly, Fig. ED3 shows vertical isotherm excursions exceeding 100m on similar time scales. The dye upwelling rate is also variable (Fig. ED7). Therefore, a steady-state assumption on the time scales of the dye release experiment (3 days) is not reasonable. It seems that the authors actually agree with this, given their deliberate use of vague language (“near-“, “relatively“, “somewhat“, etc.) without quantification of non-stationarity effects. Unfortunately, without the steady-state assumption, comparisons with the long-term average of the upwelling (Naveira Garabato et al. manuscript), let alone the global average (“~10,000 times higher than the global average”, ln. 8) are not meaningful.

Overall, it seems more appropriate to reframe the observations as those of a significant, but potentially episodic, localized, and/or reversible event – rather than evidence of “globally-significant upwelling” (ln. 13) or a “paradigm shift” (rebuttal). Observing boundary upwelling at these depths remains a significant accomplishment, even if the observed results may not be as robust or generalizable as one would hope. A few relatively minor issues were also introduced in this revision, see detailed comments below.

Detailed comments:

55: Avoiding the BBL term does not avoid the issue of dynamical partitioning between the strongly-upwelling BBL, strongly-downwelling SML, and weakly-downwelling interior discussed in theoretical literature (e.g., McDougall and Ferrari, 2017).

68: “associated with internal tides” – why are internal (as opposed to barotropic) tides singled out here? I don’t think this distinction can be made from the observations presented (nor does it matter much).

70: Fig. ED2 needs more context. Specifically, the use of TPXO tidal model needs explanation: Why barotropic tide is plotted when it is claimed (ln. 68) that the dominant mixing is due to internal tides? By how much did it need to be scaled?

72: The “heaving the time-mean profile” picture is strikingly one-dimensional and it is at odds with the (correct) assertion that “1-d treatments of the boundary are not sufficient” (ln. 173). All the transects shown demonstrate that substantial lateral gradients are present and, presumably, advected by the tides.

80: “Although the direction of flow is different, the sign of the shear remains the same” – how can this be possible? Is the flow no-slip during one phase of the tide and bottom-intensified during the reversal?

I don't see evidence of it in the velocity plots, nor does it match the Fig. 4 illustration in the Naveira Garabato (in prep.) paper.

83: Here, the discussion alternates between the "convective instability" mechanism (presented as the main driver of observed boundary mixing) and the internal tide breaking. To me, the two mechanisms are inherently different with respect to their mechanics (the former is frictional, the latter is not) as well as their localization relative to the bottom (BBL vs. interior). The relationship between the two mechanisms needs to be clarified in text. In particular, I'd like to see how the slope criticality parameter enters the convective (in-)stability analysis.

92: "relatively time-invariant temperature field" – I don't find this statement very convincing – it is vague ("relatively" to what?) and does not match the evidence presented: Fig. ED3 shows >100m isotherm excursions over time scales of 1-10 days, which is comparable to the dye experiment duration. I understand the strong motivation to describe the system as "quasi steady-state", but I'm afraid the observations do not support it. I could be convinced otherwise with some quantification of the observed scales of variability, perhaps.

109: "owing to its much higher detection threshold" – "its" is somewhat ambiguous here.

113: I don't quite see how this conclusion about the "net transformation" follows from the preceding. To me, the limited duration of dye experiments allows observation of only transient transformation, which is the opposite of "net".

142: The change from 250 ± 75 m/day estimate presented earlier to $O(100)$ m/day underscores the unfortunate lack of robustness of the results.

The considerable variability among the different estimates of the upwelling rate, ranging from less than 100 to more than 300 m/day, should be explicitly recognized within the main text, rather than being relegated to the "methods" section.

Fig. ED8: Would it be better to use a logarithmic scale for the dye concentration? Currently, the distributions in panels a-b are oversaturated, while the rest are barely visible.

220: The estimate may be "current best" estimate of upwelling of this particular patch at this particular time and place, but this does not make it a better estimate of the "upwelling occurring at the bottom boundary" in general.

485: What exactly is "uniformly distributed"? The dye distribution is certainly patchy, and so is the distribution of its center of mass.

Referee #3 (Remarks to the Author):

I thank the authors for their work in responding to the comments of myself and the other reviewers and changing the manuscript accordingly. I find that this version is much improved. In particular, the detailed discussion of the methods used to estimate the upwelling rate, the inclusion of four different estimates, the change to " $O(100\text{m/day})$ " as the headline result consistent with the four estimates and the discussion of the uncertainties involved in each now makes me much more confident in the main result. I also appreciate the new clarity around dye versus tracer, whether the system is in steady state and avoiding using the problematic and debatable phrase "bottom boundary layer" throughout most of the manuscript. I have a series of requests for minor clarifications and edits below. I am happy to

recommend publication once these minor issues have been dealt with.

I emphasize again that the results presented in this article, as the first direct (tracer/dye-based) measurement of the long-hypothesized intense near-boundary diapycnal upwelling, are very significant and worthy of wide dissemination and publication in Nature. I hope that the results will stimulate more research, in particular more observational campaigns, to understand this highly complex and important aspect of ocean circulation.

- Review response: I have one issue with several statements made in the review response. Specifically, the authors state that "i) both dye and tracer measure NET diapycnal transport". Of course this is true - they measure the diapycnal transport averaged over their extent. However, this is not the point made by myself and reviewer 1. The point is that, since the dye remains near the boundary for most of the experiment (with reference to the discussion in the first round about the ambiguity with what "near-boundary" means) the "net" diapycnal transport measured by the dye is most appropriately thought of as an estimate of just near-boundary upwelling, and so shouldn't be compared to measurements (e.g. made with long-term tracer experiments) of both near-boundary upwelling and interior downwelling averaged together. I believe the authors understand this and no such statement (the dye measures "net" upwelling) is included in the main manuscript.

- Line 77-78: Are citations to "in prep" articles allowed?

- Lines 92-94: "relatively time-invariant" and "varies somewhat in time". These are vague statements that don't really convey much meaning. The key question is whether the assumption that the vertical component of the time-mean across-canyon flow can be assumed to be a good approximation of the diapycnal velocity. Such an approximation should be able to be tested by comparing expected temperature changes due to adiabatic advection to those actually seen in Fig. ED3. Since Figure ED3 does not show a tendency toward cooling of the whole canyon over time, the result seems pretty clear but could be better described in this article. There is a statement at lines 95-97, but again "steadiness" is a relatively vague term.

- Line 93: Should this second figure reference be to ED2 rather than Figure 3? In fact, I am not sure what is meant by "superimposed on a time-mean up-canyon flow"?

- Line 114: Suggest adding "diapycnal" in front of "transformation".

- Line 126-127: I think this could be stated more simply as you are only interested in diapycnal transformation not upwelling in physical space.

- Line 130: The ED figures are referenced out of order.

- Figure ED7 caption: Weighed -> weighted.

- Line 136: Do you mean "blue contour"?
- Lines 167-168: Suggest adding "averaged over all times/transects" before "is then".
- Line 173: Suggest "1-d" -> "one-dimensional" for a general audience.
- Line 185: Why not "3-d-plus-time" -> "four-dimensional"?
- Paragraph starting at line 202: Repeating a point from my earlier comments, I think the authors could make it much clearer here that, by virtue of the averaging scale, these previous studies were really estimates of *net* upwelling, averaged over large-scale regions of both near-boundary upwelling and above-boundary downwelling. Hypsometry by definition relies on varying degrees of cancellation between near-boundary upwelling and above-boundary downwelling. The present observational estimate is better thought of as just an estimate of near-boundary upwelling (as the updated manuscript and companion shows, e.g. the figure included in the review response), and so it's not surprising that the magnitude of that upwelling is larger. This point could be made easily by, for example, adding a sentence after the first sentence of this paragraph with something like; "These previous measurements average over regions of near-boundary upwelling and above-boundary downwelling, as opposed to our more focused observations that measure just near-boundary upwelling."
- Lines 215-219: This statement is rather vague. The truth is that the scales of the measurements are so different that it doesn't really make sense to compare them.
- Line 465: Suggest adding "randomly" in front of "sub-sampled".
- Line 466: I presume you mean "ED4b, c" here rather than "ED4a,c".
- Line 467: Suggest adding "vertical" in front of "profiles".
- Line 469: "scale similarly" was not very clear to me. I think you mean that the size of the standard deviation across the different samples (Fig. ED6) varies from transect to transect in a similar way to the standard deviation of the dye-weighted density across each transect (bars in Fig. ED7). Please clarify the text.
- Figure ED7: Please add the letter labels of each transect to this figure as well for easy reference.
- Line 507: Suggest adding "(solid blue line in Fig. ED7)" after "Linear regression".
- Lines 506-507: For the time poor reader it would help to describe this weighting in words - i.e. transects with large standard deviations are weighted low.
- Line 511: "tracer" -> "dye".

- Lines 522-523: Suggest altering ". Therefore, the upwelling..." to ", yielding an upwelling..." to avoid suggesting there is a unique upwelling estimate.

- Methods section "Upwelling rate" - for presentation purposes it might be useful for the reader to put the 4 different estimates of the upwelling rate (250+-75, 101+-50, 125+-31 and 64) in a table.

Author Rebuttals to First Revision:

Response to Referee 1

In the revised manuscript, the authors addressed most of my comments and suggestions.

However, the issue of the steady-state assumption, which is critical for potential generalization of the limited observations, remains.

The authors claim that the system is “in near-steady state on timescales longer than a tidal period” and that the temperature field is “relatively time-invariant”. Figures ED2 and ED3 are presented as evidence of this claim. To me, the figures demonstrate the opposite: Upwelling velocity (Fig. ED2b) shows ample variations of 100-300 m/day on time scales of 1-10 days. Similarly, Fig. ED3 shows vertical isotherm excursions exceeding 100m on similar time scales. The dye upwelling rate is also variable (Fig. ED7).

Therefore, a steady-state assumption on the time scales of the dye release experiment (3 days) is not reasonable. It seems that the authors actually agree with this, given their deliberate use of vague language (“near-“, “relatively”, “somewhat”, etc.) without quantification of non-stationarity effects.

Unfortunately, without the steady-state assumption, comparisons with the long-term average of the upwelling (Naveira Garabato et al. manuscript), let alone the global average (“~10,000 times higher than the global average”, ln. 8) are not meaningful.

Overall, it seems more appropriate to reframe the observations as those of a significant, but potentially episodic, localized, and/or reversible event – rather than evidence of “globally-significant upwelling” (ln. 13) or a “paradigm shift” (rebuttal). Observing boundary upwelling at these depths remains a significant accomplishment, even if the observed results may not be as robust or generalizable as one would hope.

Thank you for highlighting that our previous revision lacked quantification of the steady state assumption. In line with a suggestion from Reviewer 3 we have now included an argument that includes quantification of the adiabatic vertical velocity associated with isotherm displacement and compares it to the incoming up-canyon flow and the dye-inferred diapycnal velocity. A detailed description of this argument can be found under the heading “Estimating adiabatic versus diapycnal upwelling” in the Methods section (lines 488 - 516). The adiabatic vertical velocity is significantly smaller than the inflow and the dye-inferred velocity, suggesting that the steady state assumption is valid (new Figure ED3). That said, we acknowledge that we cannot completely rule out time-dependent effects and include this caveat in the manuscript (lines 158-169 and 510-514). We have removed vague language from the manuscript, replacing the previous section on the steady-state assumption within the main body with a revised paragraph (lines 90-96) which references the new methods section and extended data item (Figure ED3).

A few relatively minor issues were also introduced in this revision, see detailed comments below.

Detailed comments:

55: Avoiding the BBL term does not avoid the issue of dynamical partitioning between the strongly-upwelling BBL, strongly-downwelling SML, and weakly-downwelling interior discussed in theoretical literature (e.g., McDougall and Ferrari, 2017).

We certainly appreciate this partitioning and have included further clarification regarding the near-boundary focus of this experiment (lines 111-113 and 205-206). Avoiding the term BBL is not intended to skirt the issue but rather to highlight that the near-boundary region we observe does not look like the well-mixed boundary layer nor have the buoyancy flux profile often seen in the theoretical literature.

68: “associated with internal tides” – why are internal (as opposed to barotropic) tides singled out here? I don’t think this distinction can be made from the observations presented (nor does it matter much).

We agree that the moored observations presented here, due to the heights of the moorings, do not demonstrate the baroclinic nature of the tides.

[REDACTED]

We also now state in the text (lines 69-72) that the kinetic energy measured is ~2-3 times larger than the barotropic tide (from TPXO, related to the next comment), again suggestive of internal tides.

[REDACTED]

70: Fig. ED2 needs more context. Specifically, the use of TPXO tidal model needs explanation: Why barotropic tide is plotted when it is claimed (ln. 68) that the dominant mixing is due to internal tides? By how much did it need to be scaled?

Thank you for pointing out the need for more detail here. We have now provided context regarding Figure ED2 and the inclusion of the TPXO comparison; specifically, that it indicates that we are observing flows due to locally generated internal tides (lines 69-72). We noticed an error in the TPXO figure in the previous version and now the time series of the TPXO velocities no longer need to be scaled. However, we highlight that TPXO kinetic energy are 2-3 times smaller than the observed kinetic energy, suggesting that we are observing flows generated by internal tides rather than the barotropic tide (lines 69-72).

72: The “heaving the time-mean profile” picture is strikingly one-dimensional and it is at odds with the (correct) assertion that “1-d treatments of the boundary are not sufficient” (In. 173). All the transects shown demonstrate that substantial lateral gradients are present and, presumably, advected by the tides.

Our choice of words here was indeed contradictory to the insufficiency of thinking about these processes in 1-d. We have now rephrased the sentence to specify that motions occur both horizontally and vertically (lines 72-73).

80: “Although the direction of flow is different, the sign of the shear remains the same” – how can this be possible? Is the flow no-slip during one phase of the tide and bottom-intensified during the reversal? I don’t see evidence of it in the velocity plots, nor does it match the Fig. 4 illustration in the Naveira Garabato (in prep.) paper.

Flow is bottom-intensified at the MP1 location, but not at the moorings further along the canyon, such as the one presented in Naveira Garabato et al. We have removed this sentence to avoid confusion as it is not central to the focus of this work.

83: Here, the discussion alternates between the “convective instability” mechanism (presented as the main driver of observed boundary mixing) and the internal tide breaking. To me, the two mechanisms are inherently different with respect to their mechanics (the former is frictional, the latter is not) as well as their localization relative to the bottom (BBL vs. interior). The relationship between the two mechanisms needs to be clarified in text. In particular, I’d like to see how the slope criticality parameter enters the convective (in-)stability analysis.

We apologize but we do not understand this comment regarding “the two mechanisms.” Our thesis is that only one instability mechanism is occurring: convective instability. When we refer to internal tide breaking, we mean convective instability. We provide evidence here, and cite Naveira Garabato et al who demonstrates it more rigorously, that this is the case. Slope criticality is presented here simply because it is an important general parameter for internal tides interacting with topography, not because a clear relationship between criticality and convective instability is known. We have attempted to clarify the text to make these points better (lines 76-79).

92: “relatively time-invariant temperature field” – I don’t find this statement very convincing – it is vague (“relatively” to what?) and does not match the evidence presented: Fig. ED3 shows >100m isotherm excursions over time scales of 1-10 days, which is comparable to the dye experiment duration. I understand the strong motivation to describe the system as “quasi steady-state”, but I’m afraid the observations do not support it. I could be convinced otherwise with some quantification of the observed scales of variability, perhaps.

As described above, we have now quantified the steady-state assumption as requested (new methods section lines 488 - 516 and Figure ED3), and have removed any vague language regarding its discussion (lines 90-96).

109: “owing to its much higher detection threshold” – “its” is somewhat ambiguous here. We have replaced “its” with “dye’s” for clarification.

113: I don’t quite see how this conclusion about the “net transformation” follows from the preceding. To me, the limited duration of dye experiments allows observation of only transient transformation, which is the opposite of “net”.

We apologise for the confusing use of terminology; we have now rephrased this sentence to clarify that the dye will measure the average transformation over the course of the experiment which may include upwelling and downwelling components depending on its extent (lines 111-113).

142: The change from 250 ± 75 m/day estimate presented earlier to $O(100)$ m/day underscores the unfortunate lack of robustness of the results.

The considerable variability among the different estimates of the upwelling rate, ranging from less than 100 to more than 300 m/day, should be explicitly recognized within the main text, rather than being relegated to the "methods" section.

We apologise, our intentions here were not to hide the range in the methods section, but hope that given that all methods results in a significant positive diapycnal velocity that some confidence can be garnered from the analysis. We now explicitly state the full range on line 143 and include a table of the values in the methods section for easy reference, as suggested by Reviewer 3.

Fig. ED8: Would it be better to use a logarithmic scale for the dye concentration? Currently, the distributions in panels a-b are oversaturated, while the rest are barely visible.

We have changed the color scale to be logarithmic for all the dye concentrations and removed some of the whiter end of the colour map (Figures 2,3,4, ED5, ED6). However, in order to distinguish regions of higher concentrations in some of the transects in the middle of the experiment the colour bar range is still relatively small and the first observation is oversaturated.

220: The estimate may be "current best" estimate of upwelling of this particular patch at this particular time and place, but this does not make it a better estimate of the "upwelling occurring at the bottom boundary" in general.

We have changed the sentence to say "Our observations provide the first direct estimate of the upwelling occurring at a bottom boundary" as we do not intend to suggest that this value is representative of the global ocean, as described in the following sentence (lines 220-223).

485: What exactly is "uniformly distributed"? The dye distribution is certainly patchy, and so is the distribution of its center of mass.

We agree that this sentence was unclear and have removed it.

Response to Referee 3

I thank the authors for their work in responding to the comments of myself and the other reviewers and changing the manuscript accordingly. I find that this version is much improved. In particular, the detailed discussion of the methods used to estimate the upwelling rate, the inclusion of four different estimates, the change to "O(100m/day)" as the headline result consistent with the four estimates and the discussion of the uncertainties involved in each now makes me much more confident in the main result. I also appreciate the new clarity around dye versus tracer, whether the system is in steady state and avoiding using the problematic and debatable phrase "bottom boundary layer" throughout most of the manuscript. I have a series of requests for minor clarifications and edits below. I am happy to recommend publication once these minor issues have been dealt with.

Thank you.

I emphasize again that the results presented in this article, as the first direct (tracer/dye-based) measurement of the long-hypothesized intense near-boundary diapycnal upwelling, are very significant and worthy of wide dissemination and publication in Nature. I hope that the results will stimulate more research, in particular more observational campaigns, to understand this highly complex and important aspect of ocean circulation.

Thank you.

- Review response: I have one issue with several statements made in the review response. Specifically, the authors state that "i) both dye and tracer measure NET diapycnal transport". Of course this is true - they measure the diapycnal transport averaged over their extent. However, this is not the point made by myself and reviewer 1. The point is that, since the dye remains near the boundary for most of the experiment (with reference to the discussion in the first round about the ambiguity with what "near-boundary" means) the "net" diapycnal transport measured by the dye is most appropriately thought of as an estimate of just near-boundary upwelling, and so shouldn't be compared to measurements (e.g. made with long-term tracer experiments) of both near-boundary upwelling and interior downwelling averaged together. I believe the authors understand this and no such statement (the dye measures "net" upwelling) is included in the main manuscript.

We do understand this and, following the suggestions made below, have included clarification in both the summary (lines 205-206) and in the main body (lines 112-114).

- Line 77-78: Are citations to "in prep" articles allowed?

We have kept these citations in for now and follow the guidelines described in Nature's formatting guide, but are happy to remove them if it is deemed inappropriate.

- Lines 92-94: "relatively time-invariant" and "varies somewhat in time". These are vague statements that don't really convey much meaning. The key question is whether the assumption that the vertical component of the time-mean across-canyon flow can be assumed to be a good approximation of the diapycnal velocity. Such an approximation should be able to be tested by comparing expected temperature changes due to adiabatic advection to those actually seen in Fig. ED3. Since Figure ED3 does not show a tendency toward cooling of the whole canyon over time, the result seems pretty clear but could be better described in this article. There is a statement at lines 95-97, but again "steadiness" is a relatively vague term.

Thank you for this suggestion. We have now included a similar argument that includes quantification of the adiabatic vertical velocity associated with the observed isotherm displacement and compares it to the incoming up-canyon flow and the dye-inferred diapycnal velocity (A detailed description of this argument can be found under the heading "Estimating adiabatic versus diapycnal upwelling" in the Methods section (lines 488 - 516) along with a new figure Figure ED3). The adiabatic vertical velocity is significantly smaller than the inflow and the dye-inferred velocity, suggesting that the steady state assumption is valid. Given this quantitative argument, we have now removed any vague language referring to steady-state assumption (lines 90-96).

- Line 93: Should this second figure reference be to ED2 rather than Figure 3? In fact, I am not sure what is meant by "superimposed on a time-mean up-canyon flow"?

We intended to reference the time mean profiles of velocity in Figure 4 rather than 3, this has been fixed. We have also rewritten this section (lines 90-96) and the description of the tidal currents (lines 67-73) and hope that the description of the presence of both an up-canyon mean flow and tidal currents is clearer.

- Line 114: Suggest adding "diapycnal" in front of "transformation".

Done.

- Line 126-127: I think this could be stated more simply as you are only interested in diapycnal transformation not upwelling in physical space.

We have attempted to rephrase this more simply on lines 125-127.

- Line 130: The ED figures are referenced out of order.

Thank you for noticing this error, the extended data items have been reordered to reflect the order they are referenced in the text.

- Figure ED7 caption: Weighed -> weighted.

Fixed.

- Line 136: Do you mean "blue contour"?

Yes, fixed.

- Lines 167-168: Suggest adding "averaged over all times/transects" before "is then".

Done.

- Line 173: Suggest "1-d" -> "one-dimensional" for a general audience.

Done.

- Line 185: Why not "3-d-plus-time" -> "four-dimensional"?

Done.

- Paragraph starting at line 202: Repeating a point from my earlier comments, I think the authors could make it much clearer here that, by virtue of the averaging scale, these previous studies were really estimates of *net* upwelling, averaged over large-scale regions of both near-boundary upwelling and above-boundary downwelling. Hypsometry by definition relies on varying degrees of cancellation between near-boundary upwelling and above-boundary downwelling. The present observational estimate is better thought of as just an estimate of near-boundary upwelling (as the updated manuscript and companion shows, e.g. the figure included in the review response), and so it's not surprising that the magnitude of that upwelling is larger. This point could be made easily by, for example, adding a sentence after the first sentence of this paragraph with something like; "These previous measurements average over regions of near-boundary upwelling and above-boundary downwelling, as opposed to our more focused observations that measure just near-boundary upwelling."

Thank you for the suggestion which clearly demonstrates the point, it has been included on lines 203-205.

- Lines 215-219: This statement is rather vague. The truth is that the scales of the measurements are so different that it doesn't really make sense to compare them.
We have removed this comparison.

- Line 465: Suggest adding "randomly" in front of "sub-sampled".
The subsampling was intentionally not done randomly, but was instead done as subsets of 10 profiles across the transects to replicate only sampling a small section of the dye cloud. This has been clarified in the text (lines 462-464).

- Line 466: I presume you mean "ED4b, c" here rather than "ED4a,c".
Yes, changed.

- Line 467: Suggest adding "vertical" in front of "profiles".
Done.

- Line 469: "scale similarly" was not very clear to me. I think you mean that the size of the standard deviation across the different samples (Fig. ED6) varies from transect to transect in a similar way to the standard deviation of the dye-weighted density across each transect (bars in Fig. ED7). Please clarify the text.

Yes, your interpretation is correct and the clarification has been added to the text (lines 465-467).

- Figure ED7: Please add the letter labels of each transect to this figure as well for easy reference.
Done.

- Line 507: Suggest adding "(solid blue line in Fig. ED7)" after "Linear regression".
Done.

- Lines 506-507: For the time poor reader it would help to describe this weighting in words - i.e. transects with large standard deviations are weighted low.
Done, now on lines 532-534.

- Line 511: "tracer" -> "dye".
Done.

- Lines 522-523: Suggest altering ". Therefore, the upwelling..." to ", yielding an upwelling..." to avoid suggesting there is a unique upwelling estimate.

Done.

- Methods section "Upwelling rate" - for presentation purposes it might be useful for the reader to put the 4 different estimates of the upwelling rate (250+-75, 101+-50, 125+-31 and 64) in a table.

Thank you for this suggestion; a table is now included in the methods section as Table A1.

Reviewer Reports on the Second Revision:

Referees' comments:

Referee #1 (Remarks to the Author):

I appreciate the authors' through responses to my reviews. The manuscript is mostly ready for publication, with some minor rewording suggested below.

The new subsection ("Estimating adiabatic versus diapycnal upwelling") adds important context to the observations and does demonstrate the relative weakness of adiabatic upwelling, as intended. I cannot agree that the strong temporal variability of the inflow (u_{in}) on the time scales of the experiment and its spatial variability along the canyon (shown in Fig. ED3) is consistent with the steady-state assumption. On the other hand, the figure does suggest that uncertainty arising from the lack of steady-state is around 50% of the mean and therefore acceptable for the order-of-magnitude estimate that is the main result of the study.

Perhaps this uncertainty could be discussed a bit, rather than dismissed with a caveat ("While we cannot completely rule out the importance of time variability...")? This would be more in line with the conclusion that "acknowledgement of the time-variable ... nature of near-boundary mixing processes may be essential."

Detailed comments:

93: "between the upwelling flow and flow across isopycnals" – this statement needs to be rephrased, since in the rest of the manuscript, "upwelling" seems to refer to cross-isopycnal flow (w^*).

Also, given the broad audience, some explanation of the "diapycnal transformation as upwelling" concept as well as diabatic/adiabatic flow would be helpful here (or earlier).

163: I am still somewhat uncomfortable with the use of "net" in this paragraph, because it very easy to confuse it with the "net" in phrases like "~30 Sv of net upwelling globally". I agree with Reviewer 3 in that "[the] diapycnal transport measured by the dye is most appropriately thought of as an estimate of just near-boundary upwelling, and so shouldn't be compared to measurements (e.g. made with long-term tracer experiments) of both near-boundary upwelling and interior downwelling averaged together [i.e., $_{net}$]."

491: "canyon walls at" -> "canyon walls"

499: "approximately equal to $\tan \alpha$ " – this seems to require an additional constraint of a constant cross-section of the canyon

501: "Figures ED2 and ED4 show" – perhaps ED2 and ED3?

507: The choice of the 3.7C isotherm is somewhat unfortunate, since it is not among the isotherms highlighted in the previous figures (e.g., Fig. 3).

509: "downstream" -> "upstream" or "down-canyon" (cf. Fig. 4)?

Fig. ED2: The figure labels and the caption alternate between “ $\sin \alpha$ ” and “ $\tan \alpha$ ”.

Fig. ED2: “Upwelling velocity” seems to refer to cross-isopycnal flow (w^*) in the rest of the manuscript.

Fig. ED3: The subpanels are not labeled.

Referee #3 (Remarks to the Author):

I thank the authors again for their work in responding to the comments of myself and the other reviewer. I am happy with the changes made. The additional material quantitatively addressing the steady state assumption is as convincing as the available data permits. I thus believe the article is ready for publication. The authors may want to consider the below minor points.

- Lines 125-127: "By focusing on the diapycnal..." I still feel that this statement makes the wrong point. You are not focusing on diapycnal transformation because it is easier. You are focusing on it because it is precisely the quantity of interest.

- Line 157: I suggest adding a reference to the "Estimating adiabatic versus diapycnal upwelling" section and/or Fig. ED3 at the end of this sentence.

- Lines 462-464: The authors might want to consider explaining in a little more detail how the subsampling is done, as it appears I have misunderstood on my previous reads. What exactly does "subsets of 10 vertical profiles were taken in sequence" mean? Does this mean every 10th profile is taken to form 10 new sets of profiles (each with a size 1/10th of the total transect)? To me, the normal approach would be to do something like bootstrapping where profiles are chosen randomly with replacement. Likely the differences between these choices of method will be small, so no need to do any more calculations, just clarify the text.

Author Rebuttals to Second Revision:

Response to Referee 1

I appreciate the authors' through responses to my reviews. The manuscript is mostly ready for publication, with some minor rewording suggested below.

Thank you.

The new subsection ("Estimating adiabatic versus diapycnal upwelling") adds important context to the observations and does demonstrate the relative weakness of adiabatic upwelling, as intended. I cannot agree that the strong temporal variability of the inflow (u_{in}) on the time scales of the experiment and its spatial variability along the canyon (shown in Fig. ED3) is consistent with the steady-state assumption. On the other hand, the figure does suggest that uncertainty arising from the lack of steady-state is around 50% of the mean and therefore acceptable for the order-of-magnitude estimate that is the main result of the study.

Perhaps this uncertainty could be discussed a bit, rather than dismissed with a caveat ("While we cannot completely rule out the importance of time variability...")? This would be more in line with the conclusion that "acknowledgement of the time-variable ... nature of near-boundary mixing processes may be essential."

Thank you for this comment. We are very happy that you agree that our new analysis better supports our conclusion. As you point out, it is in fact not necessary to prove the system is in steady state but rather that the adiabatic term is significantly smaller than the other two terms. We have therefore revised this section to shorten it as requested by the editor, to remove the offending caveat, and to provide more discussion. The new section more succinctly shows that the dominant balance is between inflow and diapycnal flow.

We now make it clear that this time variability (2-3 day time scales) is distinct from the tidal time variability that is crucial for the upwelling.

Detailed comments:

93: "between the upwelling flow and flow across isopycnals" – this statement needs to be rephrased, since in the rest of the manuscript, "upwelling" seems to refer to cross-isopycnal flow (w^*).

"Upwelling" has been changed to "in-flow" for clarify on what is now line 112.

Also, given the broad audience, some explanation of the "diapycnal transformation as upwelling" concept as well as diabatic/adiabatic flow would be helpful here (or earlier).

We have added clarification of the terms "adiabatic" and "diapycnal" on lines 109 and 111. To clarify that diapycnal transformation leads to upwelling lines 117-119 state "fluorescein dye was released near the seafloor in the centre of the canyon to measure the change in density of water parcels and, in turn, the diapycnal upwelling velocity."

163: I am still somewhat uncomfortable with the use of "net" in this paragraph, because it very easy to confuse it with the "net" in phrases like "~30 Sv of net upwelling globally". I agree with Reviewer 3 in that "[the] diapycnal transport measured by the dye is most appropriately thought of as an estimate of just near-boundary upwelling, and so shouldn't be compared to measurements (e.g. made with long-term tracer experiments) of both near-boundary upwelling and interior downwelling averaged together [i.e., \underline{net}]."

"Net" has been removed from this paragraph to avoid confusion. "Net" has been replaced by "average" in line 201 and removed elsewhere (line 207 and 209).

491: “canyon walls at” -> “canyon walls”
Changed (now line 625).

499: “approximately equal to $\tan \alpha$ ” – this seems to require an additional constraint of a constant cross-section of the canyon

A canyon with a rectangular cross-section and constant width is required for this relation to be true. This has been clarified on line 634.

501: “Figures ED2 and ED4 show” – perhaps ED2 and ED3?

Changed to just say “ED Figure 3 shows” (now line 647).

507: The choice of the 3.7C isotherm is somewhat unfortunate, since it is not among the isotherms highlighted in the previous figures (e.g., Fig. 3).

Figures 3 and ED5 have been edited to now show the 3.7C isotherm.

509: “downstream” -> “upstream” or “down-canyon” (cf. Fig. 4)?

This has been removed.

Fig. ED2: The figure labels and the caption alternate between “ $\sin \alpha$ ” and “ $\tan \alpha$ ”.

These inconsistencies have been fixed so that all labels say $\tan \alpha$.

Fig. ED2: “Upwelling velocity” seems to refer to cross-isopycnal flow (w^*) in the rest of the manuscript.

“Upwelling” has been changed to “in-flow” in the figure caption

Fig. ED3: The subpanels are not labeled.

Labels have been added to Figure ED3.

Response to Referee 3

I thank the authors again for their work in responding to the comments of myself and the other reviewer. I am happy with the changes made. The additional material quantitatively addressing the steady state assumption is as convincing as the available data permits. I thus believe the article is ready for publication. The authors may want to consider the below minor points.

Thank you.

- Lines 125-127: "By focusing on the diapycnal..." I still feel that this statement makes the wrong point. You are not focusing on diapycnal transformation because it is easier. You are focusing on it because it is precisely the quantity of interest.

We apologise for misunderstanding your point earlier. The sentence has been rephrased to "We are interested here in the diapycnal transformation of the dye, and not how the dye moves up and down due to the tides in physical space." and moved above to what is now line 146.

- Line 157: I suggest adding a reference to the "Estimating adiabatic versus diapycnal upwelling" section and/or Fig. ED3 at the end of this sentence.

A reference to both the Methods and Figure ED3 has been added on line 196.

- Lines 462-464: The authors might want to consider explaining in a little more detail how the subsampling is done, as it appears I have misunderstood on my previous reads. What exactly does "subsets of 10 vertical profiles were taken in sequence" mean? Does this mean every 10th profile is taken to form 10 new sets of profiles (each with a size 1/10th of the total transect)? To me, the normal approach would be to do something like bootstrapping where profiles are chosen randomly with replacement. Likely the differences between these choices of method will be small, so no need to do any more calculations, just clarify the text.

We have rephrased the text (now lines 589-592) which hopefully clarifies the method: "Sub-sampling involved systematically selecting 10 consecutive profiles along a transect. This approach ensured that the sub-samples represented a fraction of the dye patch. The centre of mass was then estimated for each sub-sample."